# DRP1 induces neuroinflammation via transcriptional regulation of NF-κB.

Yanhao Lai [1,2], Rebecca Z. Fan[1], Harry J. Brown[1,2,3], Said S. Salehe [1], Ethan K. Tieu[1,4] & Kim Tieu [1,2] ✉

Neuroinflammation is a major pathogenic mechanism underlying neurodegenerative diseases. Understanding how neuroinflammation is regulated is critical to therapeutic development. Here, we report that dynaminrelated protein 1 (DRP1), well-recognized for its role in mitochondrial fission, also functions as a transcription factor that regulates neuroinflammation. Using multiple inflammatory models, we demonstrate that upon stimulation with pro-inflammatory lipopolysaccharides (LPS), DRP1 translocates from the cytosol to the nucleus, where it binds to the promoter region of *Rela* (encoding NF-κB p65) to activate its gene products and other downstream inflammatory cytokines. Our data further reveal a significant role of the proinflammatory lipocalin-2 in the brain. In combination, this study identifies a previously unrecognized function of DRP1 in mediating neuroinflammation via the NF-κB-lipocalin-2 axis and highlights DRP1-mediated pathways as potential therapeutic targets for neurodegenerative and other inflammation-related diseases.

Neuroinflammation is a fundamental innate immune response in the central nervous system, involving the activation of glia cells, the release of inflammatory mediators (such as cytokines and chemokines), and the generation of reactive oxygen and nitrogen species[1]. The central role of chronic inflammation in neurodegenerative diseases such as Alzheimer's disease (AD), and Parkinson's disease (PD) has been well-documented[2]. Thus, targeting neuroinflammation represents one of the most promising therapeutic strategies for neurological disorders.

Activation of nuclear factor kappa B (NF-κB) is a central signaling pathway in inflammation, and this family of transcription factors has been intensively investigated as a therapeutic target for a wide range of diseases[3–5]. In mammals, the NF-κB family comprises five different members: p65 (RelA), RelB, c-Rel, p50 (NF-κB1), p52 (NF-κB2)[5,6]. The heterodimer formed by the p65 and p50 subunits is the most abundant and potent activator of gene transcription[6,7]. NF-κB can be activated by a wide variety of stimuli, including bacterial toxins such as lipopolysaccharide (LPS), oxidative stresses, toxic proteins, and inflammatory

cytokines[6]. Upon activation, NF-κB subunits assembles into homo- or hetero-dimerized transcription factor complexes which then translocate into the nucleus and bind to the promoter regions of target genes, leading to the transcription of pro-inflammatory cytokines[8,9]. To date, NF-κB is known to regulate the expression of hundreds of different genes[3,8–10], including *LCN2*[11]. Therefore, identifying factors that regulate its expression is highly critical.

*LCN2* encodes the lipocalin-2 protein, also known as neutrophil gelatinase-associated lipocalin (NGAL) and 24p3. Lipocalin-2 is a member of the lipocalin superfamily, a group of secreted proteins that play diverse physiological roles, including bacteriostasis, cell differentiation, iron binding, and immune response[12,13]. In addition, lipocalin-2 has been reported to act as a pro-inflammatory regulator in activated macrophage through NF-κB signaling[14]. Although the *LCN2* transcript is widely expressed in many tissues and organs, lipocalin-2 levels are either low or undetectable in the brain under basal conditions[15,16]. However, under pathological conditions or exposure to inflammatory stimuli, its expression and secretion are increased

[1]Department of Environmental Health Sciences, Florida International University, Miami, FL, USA. [2]Biomolecular Sciences Institute, Florida International University, Miami, FL, USA. [3]College of Arts and Sciences, Florida International University, Miami, FL, USA. [4]Present address: Neuroscience Program, University of Miami, Miami, FL, USA. ✉e-mail: ktieu@fiu.edu

drastically. Indeed, elevated levels of *LCN2* mRNA and lipocalin-2 have been detected in the post-mortem brain tissues of PD patients and an animal model of PD[17]. Consistent with its role in neuroinflammation, accumulating evidence indicates that lipocalin-2 is highly expressed in reactive astrocytes and microglia[18–20]. Collectively, these findings suggest that lipocalin-2 is neurotoxic when upregulated under pathological conditions. However, the mechanism by which dynamin-related protein 1 (DRP1) induces neuroinflammation via the NF-κB-lipocalin-2 axis has not been elucidated.

DRP1 is a member of the dynamin GTPase superfamily and is best known for its role in mitochondrial fission[21,22]. Independent laboratories have demonstrated that partial inhibition of DRP1 function is protective in experimental models of neurodegenerative diseases[22]. We recently reported that transcript and protein levels of DRP1 are highly elevated in the substantia nigra of PD patients[23], further supporting the potential involvement of this protein in PD pathogenesis. Consistent with its established role in mitochondrial fission, the protective effects of DRP1 inhibition have been largely attributed to restoring the balance in mitochondrial fission and fusion. However, additional mechanisms are involved. For example, we recently reported that partial deletion of DRP1 enhances autophagic flux independent of its role in mitochondrial fission[24]. Other investigators have reported that DRP1 predominantly resides in the cytosol, with only ~3% co-fractionating with mitochondria[25]. Using live-cell imaging of fluorescently labeled DRP1, a subsequent study corroborated that much of DRP1 is diffusely distributed in the cytoplasm and only ~2.5% of DRP1 puncta engage in mitochondrial fission[26]. Furthermore, DRP1 has been shown to colocalizes with other organelles, including lysosomes[27], the endoplasmic reticulum[28], and peroxisomes[29]. In combination, these results indicate that DRP1 is a multifunctional protein that exerts diverse cellular effects beyond mitochondrial division.

In this study, we provide evidence showing that DRP1 induces neuroinflammation through a transcriptional mechanism. Using LPS to activate microglia, we observed the translocation of DRP1 from the cytosol to the nucleus, where it binds to the promoter of the *Rela* gene, resulting in the upregulation of its gene products and other downstream inflammatory cytokines, including *Lcn2*. Among proinflammatory genes, *Lcn2* exhibited the highest inducibility and marked cell-type specificity. Elevated *Lcn2* levels were also detected in transgenic mice overexpressing human α-synuclein. These effects were significantly attenuated in mutant mice with a heterozygous DRP1 knockout. Collectively, our study identifies the DRP1-NF-κB-lipocalin-2 axis as a key pathway mediating neuroinflammation.

## Results

### Neuroinflammation in mouse models is reduced by partial DRP1-knockdown

Reduced DRP1 function has been shown to attenuate neuroinflammatory responses in LPS-treated cultured microglia[30,31]. However, the in vivo relevance of these observations and the mechanism by which DRP1 induces neuroinflammation requires further investigation. To address these gaps, we used our recently generated *Dnm1l*[+/−] mice, which exhibit partial DRP1 deficiency[24]. *Dnm1l*[+/−] mice and their wild-type (WT, *Dnm1l*[+/+]) littermates were injected intraperitoneally with a single dose of LPS (5 mg/kg)[32,33], and their ventral midbrain (VMB) tissue was collected 6 h later for RNA analysis. Taking an unbiased approach, we performed NanoString nCounter gene expression analysis using the Mouse Neuroinflammation Panel, which profiles 757 genes across major inflammatory pathways. Consistent with the proinflammatory effects of LPS, we observed robust dysregulation of proinflammatory transcripts in WT mice following LPS administration. In contrast, these responses were significantly attenuated in *Dnm1l*[+/−] littermates (Fig. 1a; Supplementary Data 1). The top 10 differentially expressed genes are shown in Fig. 1b, with *Lcn2* exhibiting the most dramatic induction in LPS-treated WT mice (Fig. 1b). *Lcn2* mRNA levels

reached $1510 \pm 209.5$ counts in *Dnm1l*[+/+] mice, compared to $543.6 \pm 199.9$ in *Dnm1l*[+/−] mice, a significant attenuation ($p = 0.0005$; Fig. 1c). No sex difference was detectable, and data were combined.

To determine whether DRP1 reduction protects against *Lcn2* elevation beyond the LPS model, we crossed *Dnm1l*[+/−] mice[24] with transgenic mice overexpressing human α-synuclein. At 6 months of age, *SNCA*[+/−] mice exhibited a 3.9-fold increase in *Lcn2* expression relative to WT littermates; however, this elevation was fully normalized in double-mutant *Dnm1l*[+/−]:*SNCA*[+/−] mice (Fig. 1d). Together, these results demonstrate that partial DRP1 inhibition effectively reduces neuroinflammatory gene expression, particularly *Lcn2*.

### Reduced proinflammatory gene expression in LPS-treated *Dnm1l*[+/−] mice

To validate the NanoString data and determine whether changes in *Lcn2* expression correlate with alterations in other proinflammatory genes, we quantified mRNA levels of selected inflammatory markers in the VMB of LPS-treated mice. These genes included those known to be regulated by *Lcn2*[14], such as *Nlrp3*, *Il1b*, as well as genes reported to regulate *Lcn2* expression[34,35], including *Rela*, *Il6*, *Tnfα*. All these transcripts were significantly upregulated in LPS-treated *Dnm1l*[+/+] mice, but their induction was markedly attenuated in *Dnm1l*[+/−] littermates (Fig. 2a), mirroring the pattern observed for *Lcn2* (Fig. 1a–c).

To corroborate the gene expression data obtained from the NanoString and qPCR analysis, we next quantified proinflammatory protein levels using MSD multiplex immunoassay in VMB lysates from LPS-treated *Dnm1l*[+/+] and *Dnm1l*[+/−] mice. Consistent with the mRNA data, LPS robustly increased the abundance of several inflammatory proteins, including lipocalin-2, in *Dnm1l*[+/+] mice, whereas these increases were substantially attenuated in *Dnm1l*[+/−] mice (Fig. 2b). The increases in *Lcn2*, *Cxcl10*, and *Ccl2* were validated at the protein levels. Interestingly, additional proteins were detectable but not their transcripts. We believe this is due to key differences between transcript-level detection (NanoString) and protein-level detection (MSD multiplex ELISA immunoassay). For example, mRNA can be degraded rapidly while the protein remains stable. NanoString is highly specific, but it has a narrow dynamic range of mRNA abundance, whereas MSD uses electrochemiluminescence and can detect very low femtogram-level protein concentrations. Last, there are temporal differences in gene vs protein expression. Changes in mRNA often occur earlier and transiently, whereas proteins may accumulate or persist longer. Morphologically, microglia in the *Dnm1l*[+/−] animals also appeared less activated (more ramified structures) as quantified using Sholl analysis of IBA-1 immunostaining (Supplementary Fig. 1).

Nuclear factor erythroid 2-related factor 2 (NRF2) is a transcription factor known for its antioxidant and cytoprotective functions and has been reported to modulate NF-κB−mediated inflammatory responses. Loss of NRF2 enhances inflammation by promoting NF-κB activation and the expression of downstream pro-inflammatory cytokines, whereas pharmacological or genetic activation of NRF2 suppresses NF-κB−driven inflammation[36–38]. Collectively, these studies suggest that NRF2 serves as a negative regulator of NF-κB. To determine whether NRF2 contributes to NF-κB activity in our model, we analyzed our NanoString dataset and found that *Nfe2l2* (the gene encoding NRF2) mRNA levels in the VMB were not significantly changed following LPS treatment (Supplementary Fig. 2a). We further validated this observation by qPCR, which similarly showed no alteration in *Nfe2l2* expression (Supplementary Fig. 2b). Together, these results indicate that NRF2 is unlikely to contribute to the neuroinflammatory response observed in this study.

### Expression of proinflammatory genes and cell-type−specific inducibility of *Lcn2*

Given that gene expression analyses from whole VMB tissue (Figs. 1 and 2) do not reveal the cellular sources of *Lcn2*, and because

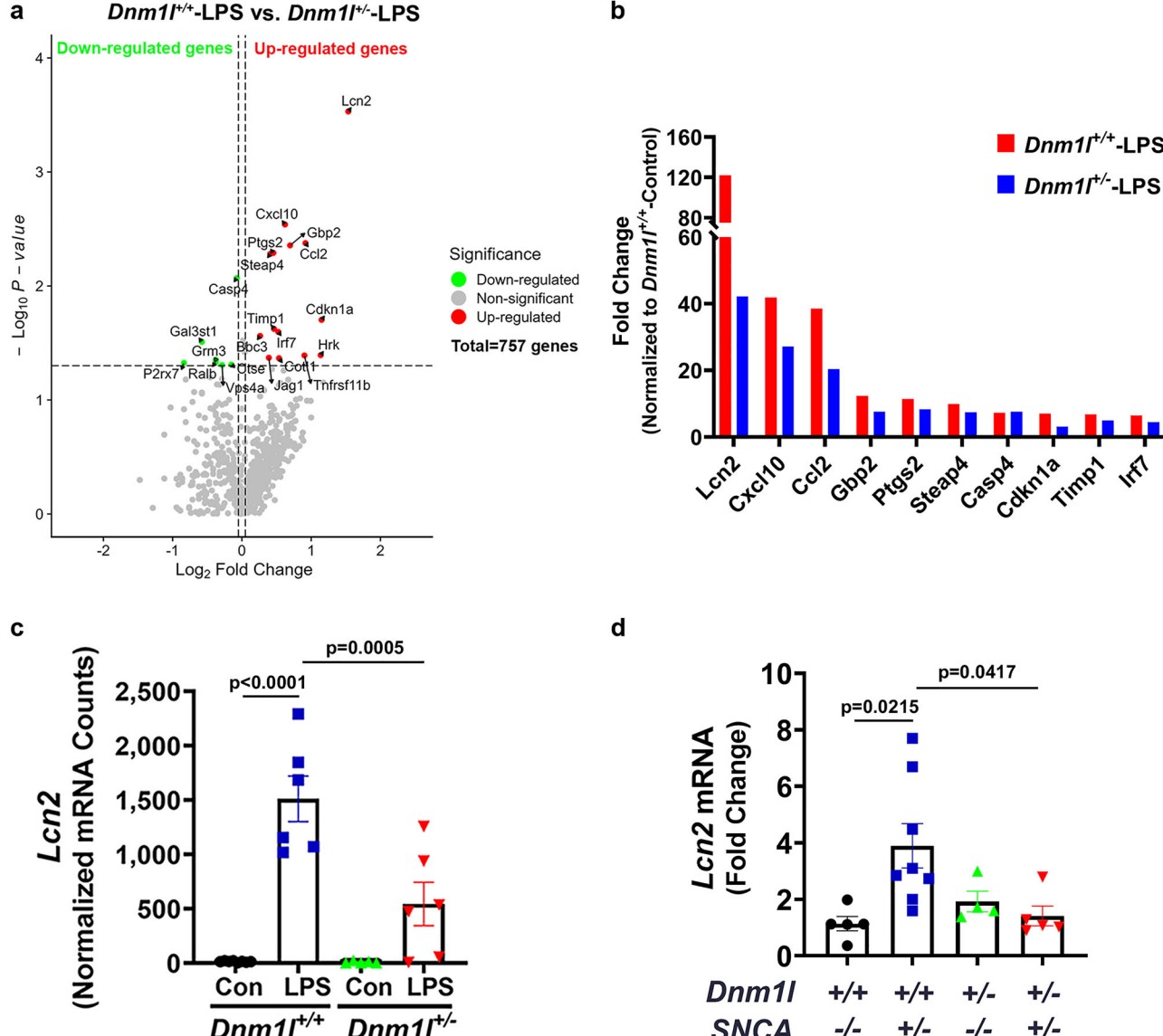

**Fig. 1 | Differential gene expression in the mouse ventral midbrain. a–c** Three-month-old *Dnm1l*$^{+/-}$ mice and their *Dnm1l* + /+ littermates received a single intra-peritoneal injection of LPS (5 mg/kg) or saline (control). Ventral midbrains were collected 6 h later for NanoString nCounter gene expression analysis (N = 5–8 mice per group; 2–4 F & 3–4 M). **a** Volcano plot depicting differentially expressed genes (DEGs) between LPS-treated *Dnm1l*$^{+/+}$ and *Dnm1l*$^{+/-}$ mice (threshold: 0.05-fold change, *p* < 0.05). **b** Top 10 DEGs identified by NanoString analysis. **c** Quantification of *Lcn2* mRNA levels, normalized using *Aars*, *Ccdc127*, *Cnot10*, *Tada2b*, and *Xpnpep1* as reference genes. N = 5–8 mice per group (2–4 F & 3–4 M); Data are presented as mean ± SEM, two-way ANOVA. **d** qPCR analysis of *Lcn2* expression (normalized to *Gapdh*) in the VMB of 6-month-old mice generated by crossing *Dnm1l*$^{+/-}$ with *SNCA*$^{+/-}$ mice (N = 4–8 mice per group; 2–4 F & 1–5 M; Data are presented as mean ± SEM, one-way ANOVA. Tukey's post hoc test was applied following all ANOVA analyses. Source data are provided as a Source Data file.

cell-type–specific expression of *Lcn2* in vivo has not been comprehensively evaluated, we used laser microdissection to isolate individual microglia, astrocytes, nigral dopaminergic (DA) neurons, and gamma-aminobutyric acid (GABA) neurons from the substantia nigra of LPS-treated *Dnm1l*$^{+/+}$ and *Dnm1l*$^{+/-}$ mice. As shown in Fig. 3a, *Lcn2* was constitutively expressed in all four cell types, with no detectable differences in basal expression across genotypes or cell populations (Fig. 3b). Following LPS exposure, *Lcn2* expression was significantly increased in microglia, astrocytes, and DA neurons of *Dnm1l*$^{+/+}$ mice, but not in GABA neurons (Fig. 3a). Consistent with whole-tissue findings (Fig. 1b), this LPS-induced upregulation was absent in microglia, astrocytes, and DA neurons microdissected from *Dnm1l*$^{+/-}$ mice (Fig. 3a).

To compare *Lcn2* inducibility across cell types, *Lcn2* expression in each population was normalized to basal *Lcn2* levels in microglia from

untreated *Dnm1l*$^{+/+}$ mice. These analyses revealed that microglia and astrocytes exhibit the highest inducibility of *Lcn2* following LPS stimulation, whereas DA neurons showed a more modest response and GABA neurons showed none (Fig. 3b). Overall, these data suggest that the increases in *Lcn2* observed in whole VMB tissue (Fig. 1) arise primarily from glial populations, particularly microglia and astrocytes, and to a lesser extent from DA neurons.

To complement the results obtained from the microdissected cells in the global heterozygous DRP1-deficient (*Dnm1l*$^{+/-}$) model (Fig. 3a, b), we next examined mice with cell-type–specific conditional deletion of *Dnm1l*. Our recently generated floxed-*Dnm1l* mice were crossed with *Cx3Cr1*$^{CreERT2}$ or *Aldh1L1*$^{CreERT2}$ lines to generate microglia-specific ("Microglia-KO") or astrocyte-specific ("Astrocyte-KO") *Dnm1l* knockdown mice, respectively. Tamoxifen-treated animals were analyzed 3 weeks later. Laser microdissection followed by Smart-seq2/

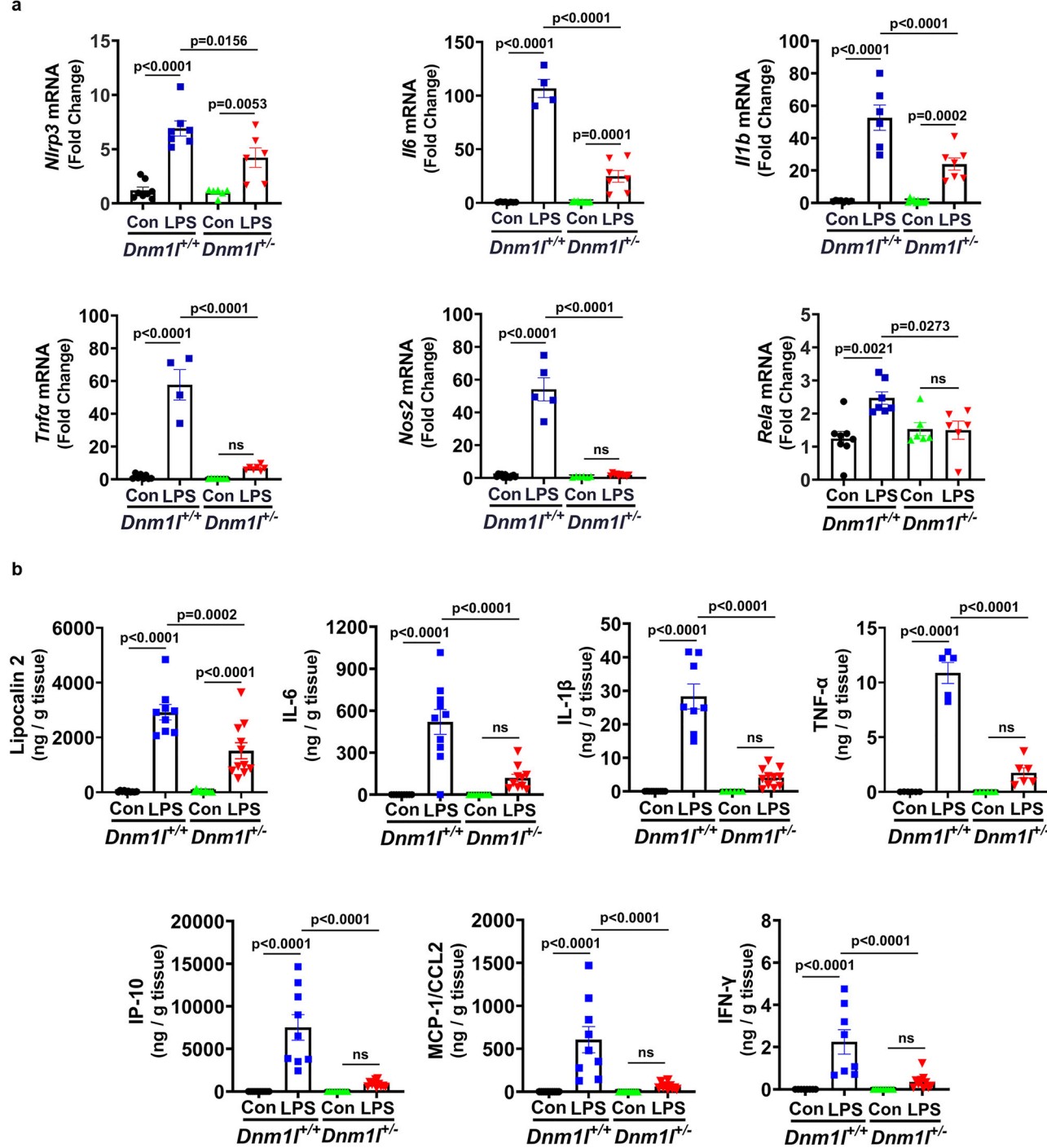

**Fig. 2 | Reduced proinflammatory gene expression in LPS-treated *Dnm1l*[+/−] mice.** Three-month-old *Dnm1l*[+/−] mice and their *Dnm1l*[+/+] littermates received a single injection of LPS (5 mg/kg, i.p.) or saline. Ventral midbrains were collected 6 h later for qPCR analysis (**a**) and MSD multiplex immunoassay (**b**). **a** Relative mRNA expression of *Nlrp3*, *Il6*, *Il1b*, *Tnfα*, *Nos2*, and *Rela*, normalized to *Gapdh*. N = 4–10 mice per group (2–5 F & 1–5 M); Data are presented as mean ± SEM, two-way

ANOVA. **b** Protein levels of proinflammatory cytokines measured by MSD multiplex immunoassay, normalized to total protein concentration. N = 10–11 mice per group (5–6 F & 5–6 M); Data are presented as mean ± SEM, two-way ANOVA. Tukey's post hoc test was applied following all ANOVA analyses. Source data are provided as a Source Data file.

qPCR confirmed a 32% reduction of *Dnm1l* in microglia of Microglia-KO mice and a 44% reduction in astrocytes of Astrocyte-KO mice (Supplementary Fig. 3). Importantly, *Dnm1l* levels were unchanged in non-targeted cell populations, demonstrating successful cell-type specificity.

Consistent with observations from the global *Dnm1l*[+/−] mice, LPS significantly increased expression of proinflammatory genes,

including *Lcn2*, *Il6*, *Il1b*, and *Tnfα*, in microglia and astrocytes from control floxed (*Dnm1l-Loxp*[+/−]) mice, relative to untreated littermates (Fig. 3c, d). However, this induction was abolished in Microglia-KO and Astrocyte-KO mice (Fig. 3c, d), indicating that partial loss of *Dnm1l* in either microglia or astrocytes is sufficient to blunt LPS-induced inflammatory gene expression. Together, these results demonstrate that both microglial and astrocytic DRP1 contribute significantly to the

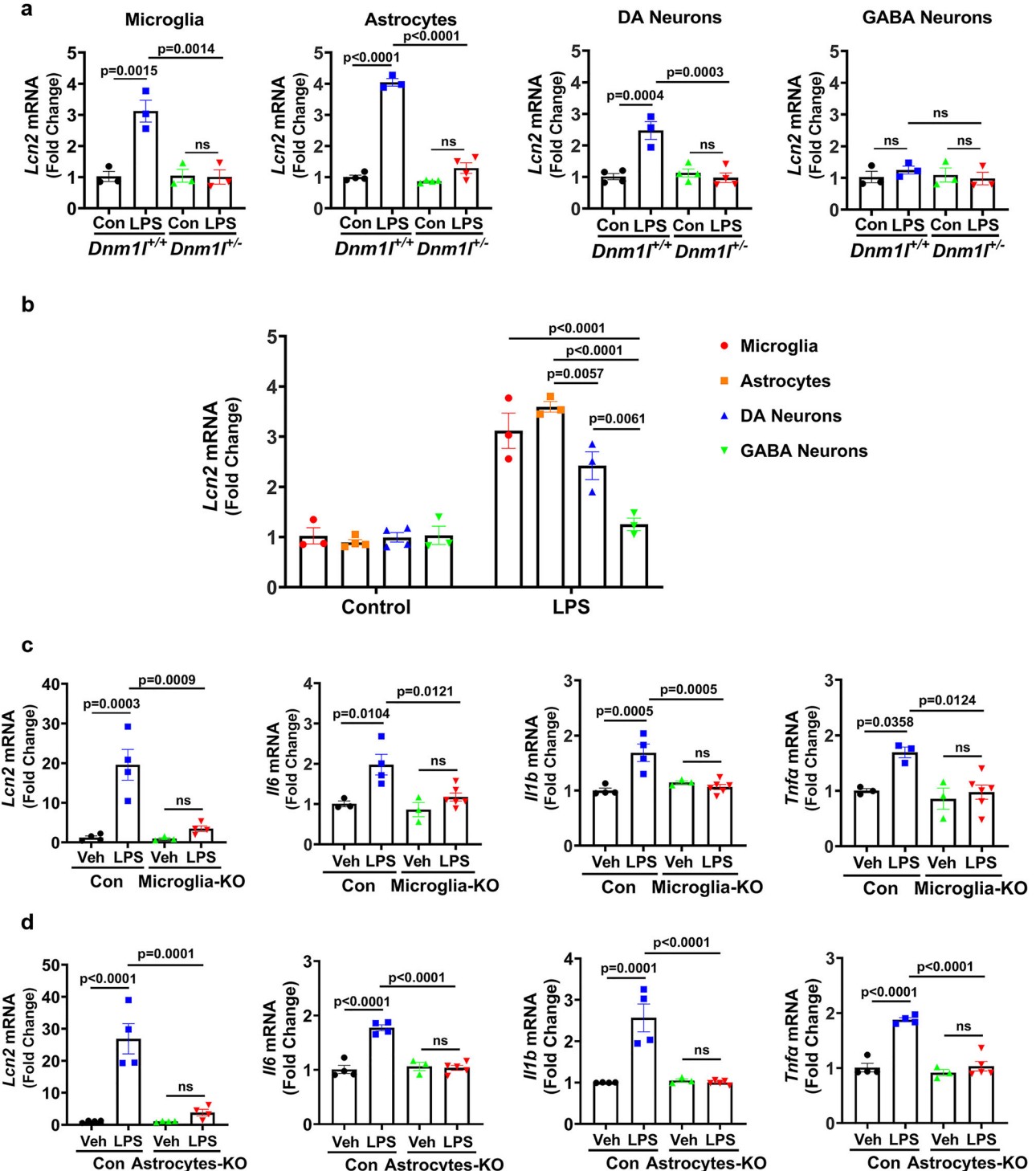

**Fig. 3 | Cell-type–specific expression of proinflammatory genes and *Lcn2* inducibility in *Dnm1l* mutant mice. a, b** Three-month-old *Dnm1l*[+/−] mice were treated with LPS (5 mg/kg, i.p.) or saline, and brains were collected 6 h later. Ventral midbrains were snap-frozen, coronally sectioned, and immunostained to identify microglia (IBA1), astrocytes (GFAP), DA neurons (TH), and GABA neurons (GAD67). Individual cells were isolated using laser microdissection. **a** Forty cells of each type per animal were microdissected from the substantia nigra, processed for cDNA, pre-amplified using the Smart-seq2 method, and analyzed by qPCR for *Lcn2* expression (normalized to *Gapdh*). N = 3–4 mice per group (2 F & 1–2 M), and data are presented as mean ± SEM. **b** Inducibility of *Lcn2* in response to LPS across cell

types in WT (*Dnm1l*[+/+]) mice. N = 3–4 mice per group (2 F & 1–2 M), and data are presented as mean ± SEM. **c, d** Tamoxifen-treated 3-month-old *Cx3Cr1-Cre*[ERT2+/−]:*Dnm1l-Loxp*[+/−] (denoted as "Microglia-KO"), *Aldh1L1-Cre*[ERT2+/−]:*Dnm1l-Loxp* + /− ("Astrocyte-KO"), and *Dnm1l-Loxp*[+/−] control mice were injected with LPS (5 mg/kg, i.p.). Brains were collected 6 h later, and microglia and astrocytes were isolated via laser microdissection. Forty cells of each type per animal were processed using Smart-seq2 and qPCR analysis of *Lcn2*, *Il6*, *Il1b*, and *Tnfα* in microglia and astrocytes (normalized to *Gapdh*). N = 3–6 mice per group (1–2 F & 1–4 M), and data are presented as mean ± SEM. Two-way ANOVA followed by Tukey's post hoc test was used for all analyses. Source data are provided as a Source Data file.

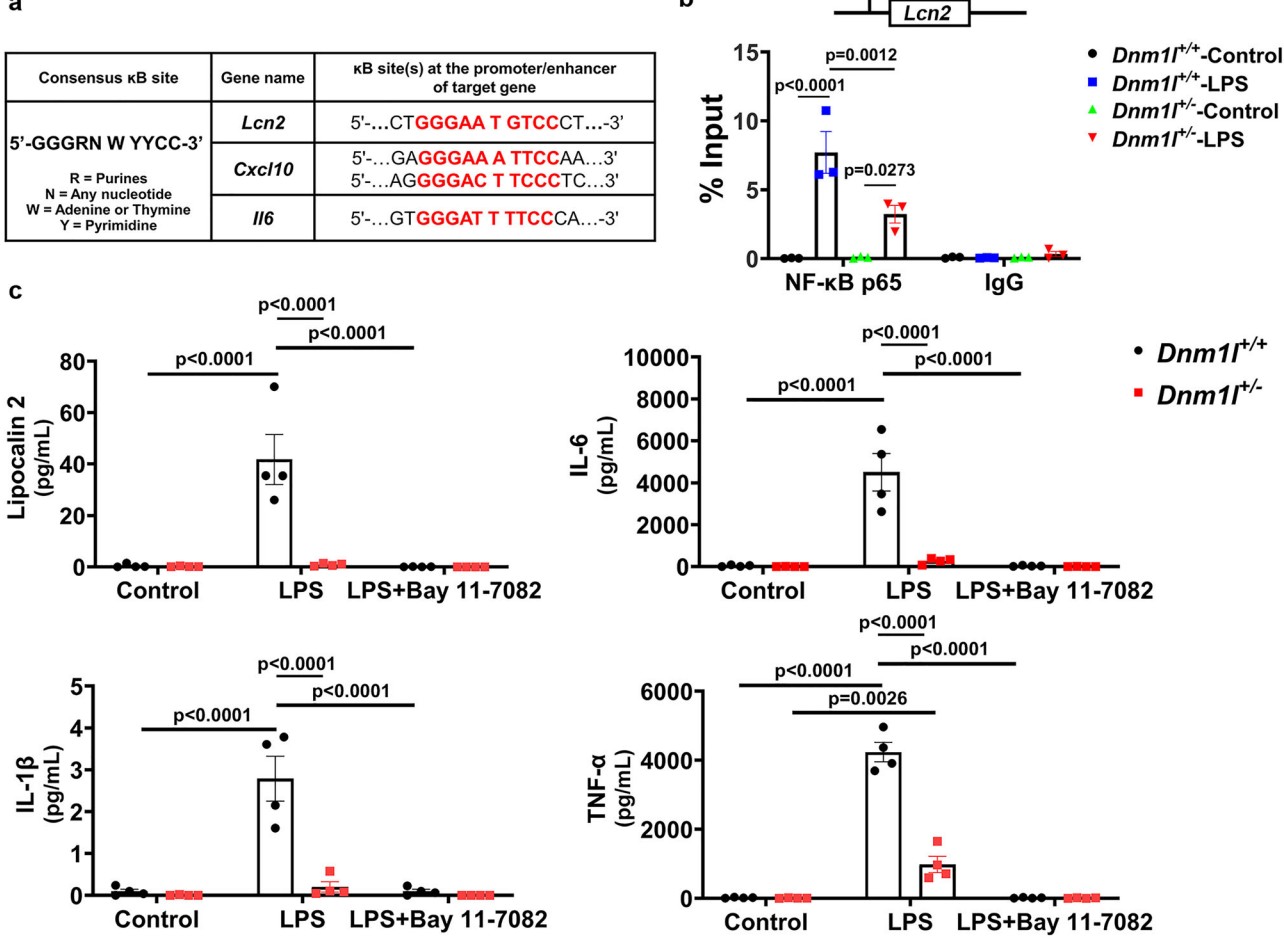

**Fig. 4 | DRP1 inhibition prevents neuroinflammation via the NF-κB pathway.**
**a** Consensus NF-κB binding sequence ("κB site") and schematic representation of κB elements located within the promoters of *Lcn2*, *Cxcl10*, and *Il6*. **b** Primary microglia from *Dnm1l*[+/+] and *Dnm1l*[+/−] mice were treated with LPS (100 ng/ml, 6 h) and subjected to ChIP analysis using rabbit anti−NF-κB p65 antibody or rabbit IgG control. Immunoprecipitated DNA ("NF-κB p65" or "IgG") was analyzed by qPCR using primers targeting the κB site in the *Lcn2* promoter. The Input % value reflects the enrichment of NF-κB p65 (relative to IgG control) at the promoter region. Data represent mean ± SEM from three independent experiments. Three-way ANOVA followed by Tukey's post hoc test. **c** Primary microglia from *Dnm1l*[+/+] and *Dnm1l*[+/−] mice were treated with LPS (100 ng/ml, 6 h) in the presence or absence of Bay11-7082 (5 μM), an NF-κB inhibitor. Conditioned media were collected for quantification of secreted cytokines by MSD multiplex immunoassay. Data represent mean ± SEM from four independent experiments. Two-way ANOVA with Tukey's post hoc test. Source data are provided as a Source Data file.

neuroinflammatory response induced by LPS, and that reducing DRP1 in either cell type effectively limits the upregulation of key inflammatory mediators such as *Lcn2*.

## DRP1 inhibition prevents neuroinflammation via the NF-κB pathway

Based on our results showing that partial DRP1 deficiency protects against LPS-induced neuroinflammation, and considering DRP1's well-established role in mitochondrial function, we first examined whether this protection might stem from preventing LPS-induced mitochondrial dysfunction. To test this possibility, we isolated microglia from adult mice treated with LPS and measured mitochondrial respiration. No mitochondrial impairment was detected under these conditions (Supplementary Fig. 4a−b). These results indicate that the reduced neuroinflammatory response observed in *Dnm1l*[+/−] mice (Figs. 1−3) is unlikely to be mediated through preservation of mitochondrial function.

To further investigate how DRP1 inhibition reduces LPS-induced neuroinflammation and microglial activation, we performed a chromatin immunoprecipitation (ChIP) assay in primary microglia treated

with LPS for 6 h, the same time point used in our in vivo studies. Because *Lcn2* has been reported to be transcriptionally regulated by NF-κB[14], and its promoter contains a κB binding motif (Fig. 4a), we examined whether NF-κB p65 is recruited to the *Lcn2* promoter following LPS exposure. In WT microglia, LPS robustly increased p65 binding to the *Lcn2* promoter (Fig. 4b). Strikingly, this enrichment was significantly reduced in microglia from *Dnm1l*[+/−] mice (Fig. 4b), indicating that DRP1 reduction reduces NF-κB recruitment to at least one of its key proinflammatory targets.

To further test the involvement of NF-κB, we measured proinflammatory cytokines in the conditioned media of LPS-treated microglia in the presence or absence of the NF-κB inhibitor Bay11-7082[39]. Pharmacological inhibition of NF-κB markedly reduced cytokine release in WT microglia (Fig. 4c). Notably, microglia from *Dnm1l*[+/−] mice displayed a similarly reduced inflammatory response (Fig. 4c). These in vitro observations are consistent with our in vivo findings showing decreased proinflammatory protein levels in the VMB of LPS-treated *Dnm1l*[+/−] mice (Fig. 2b). Together, these results reinforce a model in which DRP1 promotes neuroinflammation through activation of the NF-κB pathway.

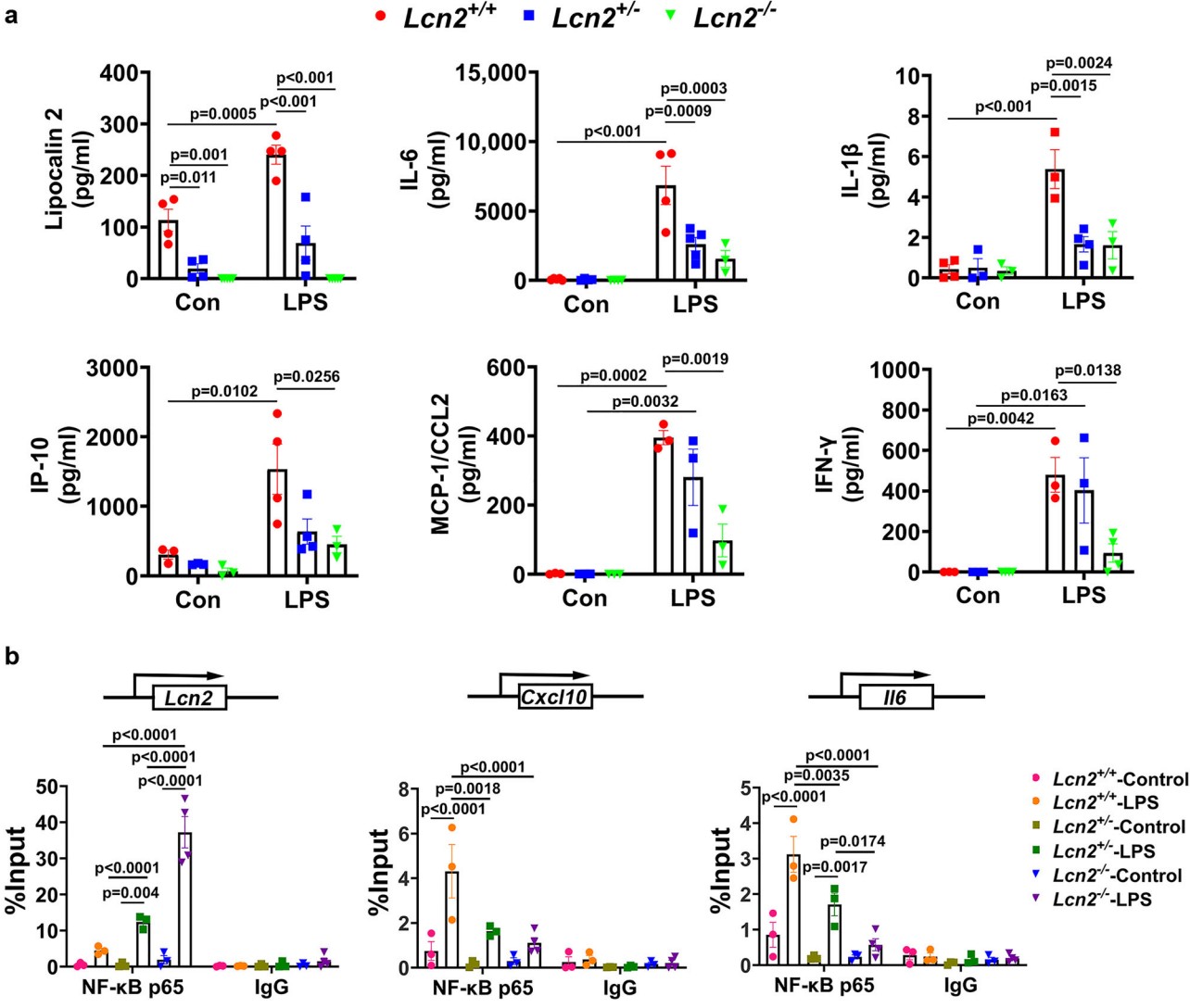

**Fig. 5 | *Lcn2* deletion reduces proinflammatory cytokine release from microglia following LPS stimulation. a** Primary microglia were plated in 96-well plates and treated with LPS (100 ng/ml) for 6 h. Conditioned media were collected and analyzed using MSD multiplex immunoassay. N = 3–5 independent experiments per analyte, with 6 technical replicates per experiment; Data are presented as mean ± SEM, two-way ANOVA followed by Tukey's post hoc test. **b** Microglia derived from *Lcn2*+/+, *Lcn2*+/−, and *Lcn2*−/− mice were treated with LPS (100 ng/ml, 6 h) and subjected to ChIP analysis as described. DNA immunoprecipitated with anti–NF-κB p65 antibody or IgG control ("NF-κB p65" or "IgG") was analyzed by qPCR using primers targeting κB sites within the promoters of *Lcn2*, *Cxcl10*, and *Il6*. Ct values were used for quantitative analysis. Data represent mean ± SEM from 3 independent experiments, three-way ANOVA followed by Tukey's post hoc test. Source data are provided as a Source Data file.

To assess whether mitochondrial alterations contribute to LPS-induced responses in primary microglia, we performed time-course studies. At the 6 h time point, the stage at which neuroinflammation is clearly induced, WT microglia exhibited neither mitochondrial fragmentation (Supplementary Fig. 4c, d) nor mitochondrial dysfunction (Supplementary Fig. 4e–h). Even at later time points (24 h), mitochondrial respiration remained unaffected. At an earlier time point (2 h), LPS transiently induced shorter mitochondria in both *Dnm1l*+/+ and *Dnm1l*+/− microglia, but this morphological change resolved without detectable functional deficits. Collectively, these results demonstrate three key points: First, LPS activates microglia without inducing mitochondrial dysfunction under the experimental conditions used. Second, early, transient mitochondrial morphological changes occur but are reversible. Third, the protective effects of partial DRP1 deficiency against neuroinflammation occur through a mechanism independent of mitochondrial impairment, instead acting through decreased NF-κB activation and target gene regulation.

## *Lcn2* deletion reduces proinflammatory cytokine release from microglia

Lipocalin-2 is increasingly recognized as a potent inducer of neuroinflammation[12,14,40]. Our data showing DRP1 deficiency reduces both NF-κB activity and lipocalin-2 levels suggest that a DRP1–NF-κB–lipocalin-2 axis contributes to the inflammatory response. To determine whether *Lcn2* expression correlates with the production of proinflammatory cytokines, primary microglia from *Lcn2*+/+, *Lcn2*+/−, and *Lcn2*−/− mice were treated with LPS, and cytokine levels in the conditioned medium were measured by multiplex ELISA. As shown in Fig. 5a, basal secretion of lipocalin-2 in *Lcn2*+/+ microglia was 5.8-fold higher than in *Lcn2*+/− microglia (p = 0.011) and undetectable in *Lcn2*−/− cells. Such genotypic differences were not observed in the basal levels of IL-6, IL-1β, CXCL-10/IP10, MCP-1/CCL2, and IFN-γ (Fig. 5a), indicating that *Lcn2* deletion selectively affects *Lcn2* itself rather than general cytokine production. Upon LPS treatment, proinflammatory cytokine secretion increased markedly in *Lcn2*+/+ microglia. *Lcn2* deletion attenuated these responses in a gene-dose–dependent manner (Fig. 5a).

Partial loss of *Lcn2* (*Lcn2*[+/−]) significantly reduced lipocalin-2, IL-6, and IL-1β, whereas complete deletion (*Lcn2*[−/−]) produced broader suppression of additional cytokines (Fig. 5a). These data support a critical role for lipocalin-2 in amplifying microglial inflammatory responses.

Given that DRP1 deficiency reduces neuroinflammation by decreasing NF-κB recruitment to its target genes, we next asked whether the reduced inflammatory responses in *Lcn2*-KO microglia might also reflect diminished NF-κB promoter recruitment. To address this question, we performed ChIP assays to measure NF-κB p65 binding to the *Lcn2* promoter using *Lcn2*[+/+], *Lcn2*[+/−], and *Lcn2*[−/−] microglia. Surprisingly, NF-κB p65 enrichment at the *Lcn2* promoter was *inversely* correlated with *Lcn2* expression levels (Fig. 5b), being highest in *Lcn2*[−/−] cells. We hypothesized that this increased binding represents a compensatory attempt by NF-κB to upregulate *Lcn2* expression. However, because exons 2–5 of the *Lcn2* gene are deleted in the *Lcn2*[−/−] mice, NF-κB binding cannot drive *Lcn2* transcription; instead, NF-κB becomes sequestered, or "hijacked," at the nonfunctional *Lcn2* promoter. If this model is correct, sequestration of NF-κB at the *Lcn2* promoter should reduce its availability to bind other proinflammatory genes. To test this hypothesis, we examined NF-κB p65 recruitment to the promoters of *Cxcl10* and *Il6*, both canonical NF-κB targets (Fig. 4a). Consistent with our hypothesis, NF-κB binding to these promoters was markedly reduced in *Lcn2*[+/−] and *Lcn2*[−/−] microglia compared with *Lcn2*[+/+] microglia (Fig. 5b). These results indicate that the blunted inflammatory response observed in *Lcn2*-KO microglia is due, at least in part, to sequestration of NF-κB at the inactive *Lcn2* promoter, thereby limiting its recruitment to other proinflammatory genes. Together, these results identify a previously unrecognized mechanism by which loss of *Lcn2* reduces neuroinflammation.

## LPS induces upregulation and nuclear translocation of DRP1

Based on our observations that DRP1-KO reduced the expression of proinflammatory genes (Figs. 1, 2), we hypothesized that the reduced binding of NF-κB p65 to the promoter of *Lcn2* was due to lower expression of *Rela*, the gene encoding NF-κB p65. Indeed, qPCR analysis showed significantly reduced *Rela* expression in *Dnm1l*[+/−] mice compared with WT littermates (Fig. 2a). It has been reported that nuclear DRP1 accumulation in lung adenocarcinoma might contribute to hypoxia-induced drug resistance[41]. Although DRP1 is classically described as a cytosolic and mitochondrial protein[25], these observations raise the possibility that DRP1 may translocate to the nucleus under certain conditions. Together, such prior research led us to hypothesize that DRP1 might directly regulate transcription of proinflammatory genes.

To test this hypothesis, we used immunofluorescence and Imaris 3D reconstruction to quantify nuclear (DRP1[Nuc], red) versus cytoplasmic (DRP1[Cyto], yellow) DRP1 in primary microglia and in microglia from LPS-treated mice (Fig. 6a–d). Even in vehicle-treated controls, a low level of nuclear DRP1 was detected, an unexpected observation that, to our knowledge, has not been previously reported and it raises the question of why DRP1 is in the nucleus. Upon LPS exposure, nuclear DRP1 increased 3.8-fold in primary microglia (p = 0.0084), while cytoplasmic DRP1 increased 2.3-fold (p = 0.0001) (Fig. 6a, b). Consistent results were observed in vivo: microglia in the substantia nigra of LPS-treated mice exhibited a 3-fold increase in DRP1[Nuc] (p = 0.0009) and a 1.5-fold increase in DRP1[Cyto] (p = 0.0479) relative to vehicle-treated controls (Fig. 6c, d). These results indicate that LPS stimulation promotes both nuclear translocation of DRP1 and an overall increase in DRP1 protein levels.

To quantitatively validate these observations, we performed capillary immunoblotting on nuclear and cytosolic fractions of *Dnm1l*[+/+] and *Dnm1l*[+/−] microglia with or without LPS treatment. Fraction purity was confirmed by probing nuclear fractions for GAPDH (cytosolic marker) and cytosolic fractions for Histone H3 (nuclear marker), with no evidence of cross-contamination

(Supplementary Fig. 5). In agreement with the imaging results, LPS increased DRP1[Nuc] and DRP1[Cyto] levels by 2.5-fold (p = 0.0058) and 1.7-fold (p = 0.0072), respectively, in *Dnm1l*[+/+] microglia (Fig. 6e, f). These increases were absent in *Dnm1l*[+/−] microglia, demonstrating that partial DRP1 deficiency prevents LPS-induced DRP1 upregulation and nuclear enrichment. Similar observations were made using whole cell extracts (Supplementary Fig. 6). Collectively, these data show that LPS induces both upregulation and nuclear translocation of DRP1 in microglia, and that this process is impaired in DRP1-deficient cells. These results support a model in which nuclear DRP1 contributes to the transcriptional activation of proinflammatory genes.

## DRP1 is recruited to the promoter region of *Rela* gene upon LPS exposure

Given that *Rela* levels were lower in *Dnm1l*[+/−] mice compared with their *Dnm1l*[+/+] littermates following LPS treatment (Fig. 2a), we hypothesized that *Rela* might be transcriptionally regulated by DRP1. To test this hypothesis, we performed ChIP assays in WT microglia to determine whether DRP1 is recruited to the *Rela* promoter after LPS exposure. Because DRP1 was not a canonical transcription factor whose binding consensus motif was previously identified, we designed seven primer pairs (P1–P7) to span ~2000 bp of the *Rela* promoter (Fig. 7a; Supplementary Data 2). Strikingly, DRP1 enrichment was significantly increased at a 255 bp region amplified by primer set P7 in LPS-treated microglia compared with vehicle controls (p < 0.0001; Fig. 7b). No enrichment was detected in IgG controls, and no DRP1 recruitment was observed at regions amplified by P1–P6 (Fig. 7b). These results indicate that LPS exposure induces selective recruitment of DRP1 to a promoter region proximal to the *Rela* transcription start site.

To further validate the transcriptional regulation of *Rela* by DRP1, we cloned the 255 bp DRP1-binding region identified by ChIP into the pGL4.20[*luc2*/Puro] luciferase reporter vector (Fig. 7c). This construct, along with a promoterless control vector, was transfected into mouse neuronal cells in the presence of either *Dnm1l*-specific siRNA or a scrambled control. Untransfected cells were included as an additional control. As shown in Fig. 7d, luciferase activity in cells driven by the DRP1-binding region was markedly increased (p < 0.0001), whereas those with the promoterless vector or no transfection showed no activity. Silencing *Dnm1l* significantly reduced luciferase activity compared with the scrambled control (p = 0.0048), consistent with DRP1-dependent transcriptional activation. *Dnm1l* knockdown efficiency was 71.21 ± 0.68% (Supplementary Fig. 7). These results demonstrate that DRP1 directly regulates the *Rela* promoter in a transcriptionally active manner.

Finally, to determine the temporal relationship between DRP1 and NF-κB p65 nuclear accumulation, we performed a time-course analysis following LPS stimulation. Nuclear DRP1 increased as early as 0.5 h after LPS exposure, whereas nuclear NF-κB p65 began to increase at 3 h (Fig. 7g–i), indicating that DRP1 translocates to the nucleus prior to NF-κB p65. We did not observe a significant increase in cytosolic NF-κB p65 levels up to 6 h after LPS treatment, supporting the notion that DRP1 promotes NF-κB p65 nuclear translocation rather than retention in the cytosol. Fraction purity was confirmed by probing nuclear fractions for GAPDH and cytosolic fractions for Histone H3 (Supplementary Fig. 8).

## Discussion

Neuroinflammation is a pathological hallmark of neurodegenerative diseases, and its negative impact underscores the need to elucidate the molecular mechanisms that regulate inflammatory responses in the central nervous system. Among these mechanisms, the NF-κB pathway is one of the most extensively studied and is widely recognized as a master regulator of inflammation[9,42]. NF-κB controls the transcription of numerous proinflammatory genes[3,10], including those required for formation of the NLRP3 inflammasome, such as *Nlrp3*, *pro−Il1b*,

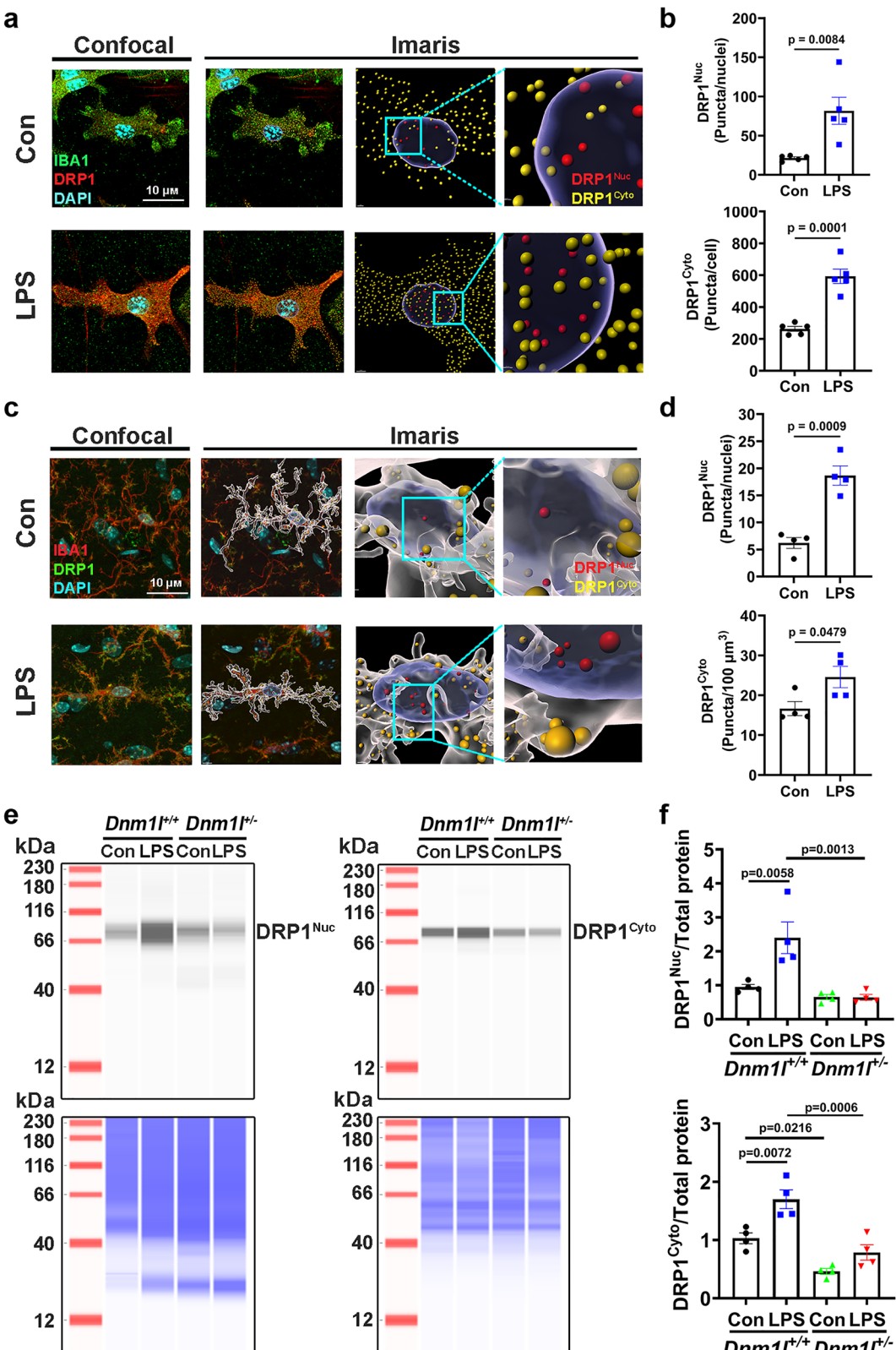

pro–Il18, pro–caspase-1, and Asc[43]. Once activated, the NLRP3 inflammasome promotes the maturation and release of potent cytokines[44]. NF-κB–NLRP3 axis represents a major inflammatory pathway.

In this study, we provide evidence that DRP1, traditionally characterized as a cytosolic and mitochondrial fission protein, also functions as a transcriptional regulator of NF-κB. We detected low levels of nuclear DRP1 under basal conditions and observed a substantial increase in nuclear DRP1 following LPS stimulation. DRP1 was recruited to a discrete region of the Rela promoter, enhancing Rela transcription and subsequently elevating downstream NF-κB target genes, including Nlrp3 and Lcn2. Lipocalin-2 has emerged as a key mediator of neuroinflammation[11,13,45]. Our results show that Lcn2 expression increased in microglia, astrocytes, and DA neurons, but not GABA neurons, following LPS exposure. The lack of induction in GABA

**Fig. 6 | LPS induces upregulation and nuclear translocation of DRP1. a, b** Primary microglia cultured from *Dnm1l*[+/+] pups were plated on chamber slides and treated with LPS (100 ng/ml) or vehicle for 6 h. **a** Representative confocal images of IBA1 and DRP1 immunofluorescence, followed by Imaris 3D rendering. **b** Quantification of nuclear DRP1 (red) and cytoplasmic DRP1 (yellow) using Imaris. N = 5 independent experiments, 14–17 cells per experiment; Data are presented as mean ± SEM, unpaired two-sided t-test. **c, d** Three-month-old *Dnm1l*[+/+] mice were injected with a single dose of LPS (5 mg/kg, i.p.) or saline. **c** Midbrain sections containing the SNpc were immunostained for microglia and DRP1, followed by Imaris 3D reconstruction. **d** Quantification of nuclear and cytosolic DRP1 puncta in SNpc microglia. N = 4 mice (2 F & 2 M), 10–15 cells per animal; Data are presented as mean ± SEM, unpaired two-sided t-test. **e, f** Primary microglia from *Dnm1l*[+/+] and *Dnm1l*[+/−] mice were treated with LPS (100 ng/ml, 6 h) and subjected to nuclear and cytoplasmic fractionation. **e** Immunoblotting of DRP1 in nuclear and cytoplasmic fractions (top panels); total protein per lane served as loading control (bottom panels). **f** Quantification of DRP1 levels in nuclear (DRP1[Nuc]) and cytoplasmic (DRP1[Cyto]) fractions. N = 4 independent experiments; Data are presented as mean ± SEM, two-way ANOVA followed by Tukey's post hoc test. Source data are provided as a Source Data file.

---

neurons suggests cell-type-specific differences in *Lcn2* inducibility which warrants future studies. Combined with the demonstration that LPS induced DRP1 translocation in microglia, we believe that the upregulation of *Lcn2* observed in the cell type is unlikely to originate from neuronal sources alone, and the elevated *Lcn2* detected in vivo likely arises predominantly from glial sources.

The link between lipocalin-2 and the NF-κB–NLRP3 inflammasome has been investigated in multiple contexts. For example, lipocalin-2 can activate the NLRP3 inflammasome via DAMP-driven TLR4 signaling in heart failure models[46], and in macrophages[14], lipocalin-2 directly enhances NLRP3 expression through NF-κB activation. Consistent with these reports, our data show that *Lcn2* deletion suppresses proinflammatory cytokine production in LPS-stimulated microglia in a gene-dose-dependent manner. Surprisingly, the anti-inflammatory effect of lipocalin-2 deletion is mediated, at least in part, through sequestration, or "hijacking", of NF-κB p65 at the *Lcn2* promoter. Due to loss of *Lcn2* exons in the knockout mice, NF-κB accumulates at a nonfunctional promoter, reducing its availability for recruitment to other proinflammatory targets such as *Cxcl10* and *Il6*. Thus, in addition to limiting NLRP3 activation, loss of *Lcn2* function indirectly dampens NF-κB signaling by restricting p65 promoter occupancy at other targets. These results highlight lipocalin-2 as a key modulator of the NF-κB–NLRP3 inflammasome pathway.

We detected elevated levels of *Lcn2* in not only LPS models but also in transgenic mice overexpressing α-synuclein, supporting the potential involvement of lipocalin-2 in PD. Supporting this idea, lipocalin-2 is increased in the substantia nigra of PD patients and MPTP-treated mice[17]. *Lcn2*-KO mice protect against MPTP-induced neurodegeneration, whereas recombinant lipocalin-2 injection into the substantia nigra induces marked dopaminergic neurodegeneration. Clinical studies further demonstrate that cerebrospinal fluid lipocalin-2 levels positively correlate with α-synuclein and may serve as a potential biomarker for PD[47]. Whether lipocalin-2 actively contributes to α-synuclein-mediated neurotoxicity remains an important question for future investigation.

To determine whether DRP1 can drive neuroinflammation independently of mitochondrial dysfunction, the LPS regimen used in our studies generated strong inflammatory responses without inducing mitochondrial impairment in microglia, either in vivo or in vitro. This is an important consideration as a previous study has shown that mitochondrial impairment and fragmentation, induced by combined treatment with LPS and nigericin for 24 h, are required for neuroinflammation in primary rodent microglia[30]. Although this represents an important mechanism, our goal was to determine whether DRP1 can drive neuroinflammation in the absence of overt mitochondrial abnormalities. The in vivo LPS regimen used in this study was selected partly based on previous studies, which show rapid and sustained microglia activation in the substantia nigra[32,33,48]. Moreover, at this treatment time-point, LPS does not cause dopaminergic neurodegeneration[32,33]. Prior studies using this regimen show that microglial activation emerges within hours of peripheral LPS injection, but dopaminergic neurodegeneration in the substantia nigra develops much later, approximately seven months to observe modest loss (~23%) and ten months to detect more pronounced

degeneration (39–47%)[32,49]. Thus, this model reflects an early, pre-degenerative stage of neuroinflammation. We believe this model represents an early stage of neuroinflammation preceding cell death in PD. This is critical to detect early translocation of DRP1 to the nucleus to initiate neuroinflammation. Additionally, we found that this paradigm markedly increased pro-inflammatory cytokines in the VMB, yet mitochondrial respiration in microglia remained intact (Supplementary Fig. 4a, b). Similarly, primary microglia treated with LPS for 6 h showed strong inflammatory responses (Fig. 4c) but no detectable changes in mitochondrial morphology or function (Supplementary Fig. 4c–h). Although transient mitochondrial fragmentation was observed at 2 h, this effect was fully reversible by 6 h and occurred without accompanying respiratory deficits. Together, these results indicate that by using neuroinflammatory models without mitochondrial dysfunction, we demonstrate that DRP1 induces neuroinflammation via a mechanism independent of mitochondria as illustrated in Fig. 8.

In addition to activating neuroinflammation, lipocalin-2 also impairs autophagy flux[46,50]. Since neuroinflammation induced by LPS also impairs autophagy[51–53], the combination of lipocalin-2 and LPS would exacerbate each other's negative impact on autophagy. One significant implication of this effect is its potential link to the accumulation of toxic proteins (such as α-synuclein, tau, and β-amyloid) and damaged organelles (mitochondria) as seen in neurodegenerative diseases. DRP1 itself is degraded via autophagy[54], and we observed LPS-induced increases in DRP1 protein levels, suggesting that impaired autophagy may further amplify DRP1's proinflammatory effects, generating a feed-forward cycle of neuroinflammation and disrupted proteostasis. Our data show that autophagy impairment induced by LPS was protected by partial loss of DRP1 function (Supplementary Fig. 9). This protection is consistent with our previous study[24]. Combined with other time-course studies, our data show that DRP1 nuclear translocation precedes NF-κB activation and subsequent p62 accumulation, supporting a model in which DRP1-driven transcriptional changes occur upstream of neuroinflammation and autophagy impairment.

To reduce DRP1 function, we used genetic approaches rather than pharmacological inhibitors, due to the concerns of the off-target effects and other limitations of DRP1 inhibitors. For example, mitochondrial division inhibitor (mdivi-1) is reported to reversibly inhibits complex I[55]. Dynasore also blocks dynamin 1 and dynamin 2[56]. Drpitor1, identified through in silico screening, exhibits antitumor activity most likely through DNA intercalation and topisomerase II inhibition[57]. P110 is a peptide-based inhibitor designed to interfere with the interaction of DRP1 with FIS1, a mitochondrial outer membrane fission protein involved in DRP1 recruitment to mitochondria[58]. This peptide also partially inhibits the GTPase activity of DRP1[58]. One limitation of using P110 is its poor oral bioavailability due to the unfavorable physiological environment (pH, enzymes) and physical barriers in the gastrointestinal tract. Indeed, in animal studies, P110-TAT is delivered through subcutaneous mini-osmotic pumps[59,60] to bypass the gastric issue. In studies where long-term treatment is required, P110 is not feasible. However, this peptide would be a great tool for in vitro or acute in vivo studies to complement the heterozygous DRP1-KO

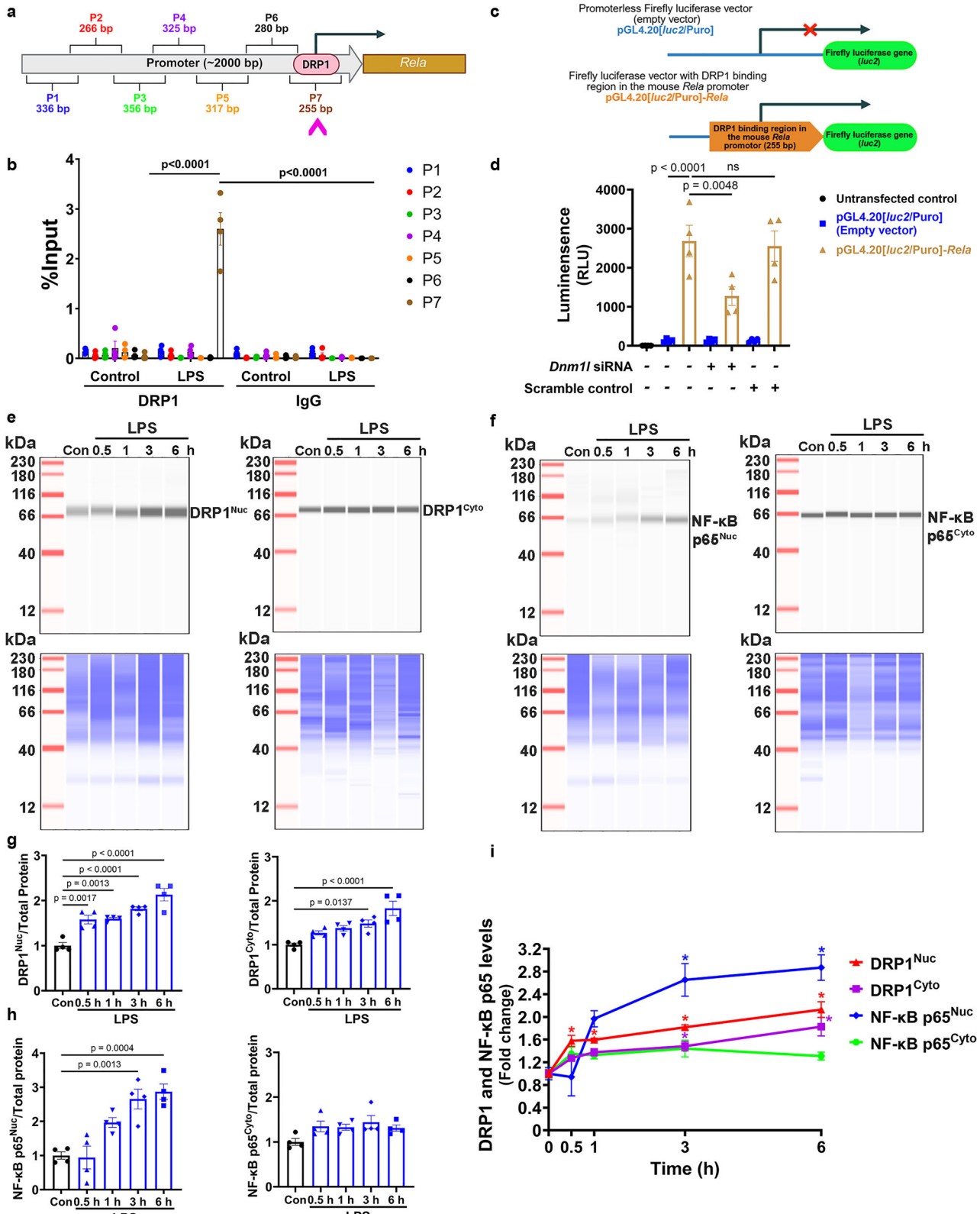

models in future studies. We used heterozygous instead of homozygous DRP1-KO mice because complete deletion of *Dnm1l* is embryonically lethal. Our *Dnm1l*[+/−] mouse model has been extensively characterized in our previous publication[24]. For example, quantitative analysis using qPCR and immunoblotting in multiple brain regions confirmed an approximately 50% reduction of DRP1 levels in the *Dnm1l*[+/−] mice. These mutant mice are indistinguishable from their WT

littermates in physical appearance, motor, and cognitive function. Stereological cell counting shows comparable numbers of DA neurons in the substantia nigra pars compacta (SNpc). Assessments of mitochondrial morphology and function in adult mice show no genotype-dependent differences. In this study, however, we performed additional experiments to characterize this mouse model. RNAseq data show that other than *Dnm1l*, all other genes are not affected in the

**Fig. 7 | DRP1 enrichment at the *Rela* promoter upon LPS exposure is transcriptionally active. a** Schematic of the -2000 bp *Rela* promoter region containing seven primer sets (P1–P7) used for ChIP analysis (Created in BioRender. Lai, Y. (2026) https://BioRender.com/m43u985). DRP1 binding was enriched at a - 255 bp region proximal to the *Rela* transcription start site following LPS treatment. **b** Primary WT microglia were treated with 100 ng/mL LPS or vehicle for 6 h, and ChIP assays were performed to assess DRP1 enrichment at the *Rela* promoter. "% input" values represent mean ± SEM from four independent experiments, two-way ANOVA with Tukey's multiple-comparison post hoc test. **c** Schematic of the pGL4.20[*luc2*/Puro]-*Rela* luciferase reporter containing the DRP1-binding region of the mouse *Rela* promoter, along with the promoterless pGL4.20[*luc2*/Puro] empty vector control (Created in BioRender. Lai, Y. (2026) https://BioRender.com/iii5iv2). **d** HT22 neuronal cells were transfected with either the pGL4.20[*luc2*/Puro]-*Rela* construct or empty vector for 24 h, in the presence or absence of *Dnm1l* siRNA or scramble control. Untransfected cells were included as an additional control. Luciferase activity was measured 48 h post-transfection. N = 4 independent experiments; Data are presented as mean ± SEM, one-way ANOVA with Tukey's post hoc test. **e–i** Primary WT microglia were treated with vehicle or 100 ng/mL LPS for 0.5, 1, 3, or 6 h. Immunoblot analysis of DRP1 (**e**) and NF-κB p65 (**f**) was performed in nuclear and cytoplasmic fractions (top panels). Total protein staining (bottom panels) served as loading control. **g** Quantification of nuclear and cytoplasmic DRP1 from (**e**). **h** Quantification of nuclear and cytoplasmic NF-κB p65 from (**f**). **i** Temporal profiles of DRP1 and NF-κB p65 enrichment in nuclear and cytosolic fractions demonstrate that DRP1 nuclear translocation precedes that of NF-κB p65. N = 4 independent experiments; Data are presented as mean ± SEM, one-way ANOVA with Tukey's post hoc test. "*" denotes p < 0.05. Source data are provided as a Source Data file.

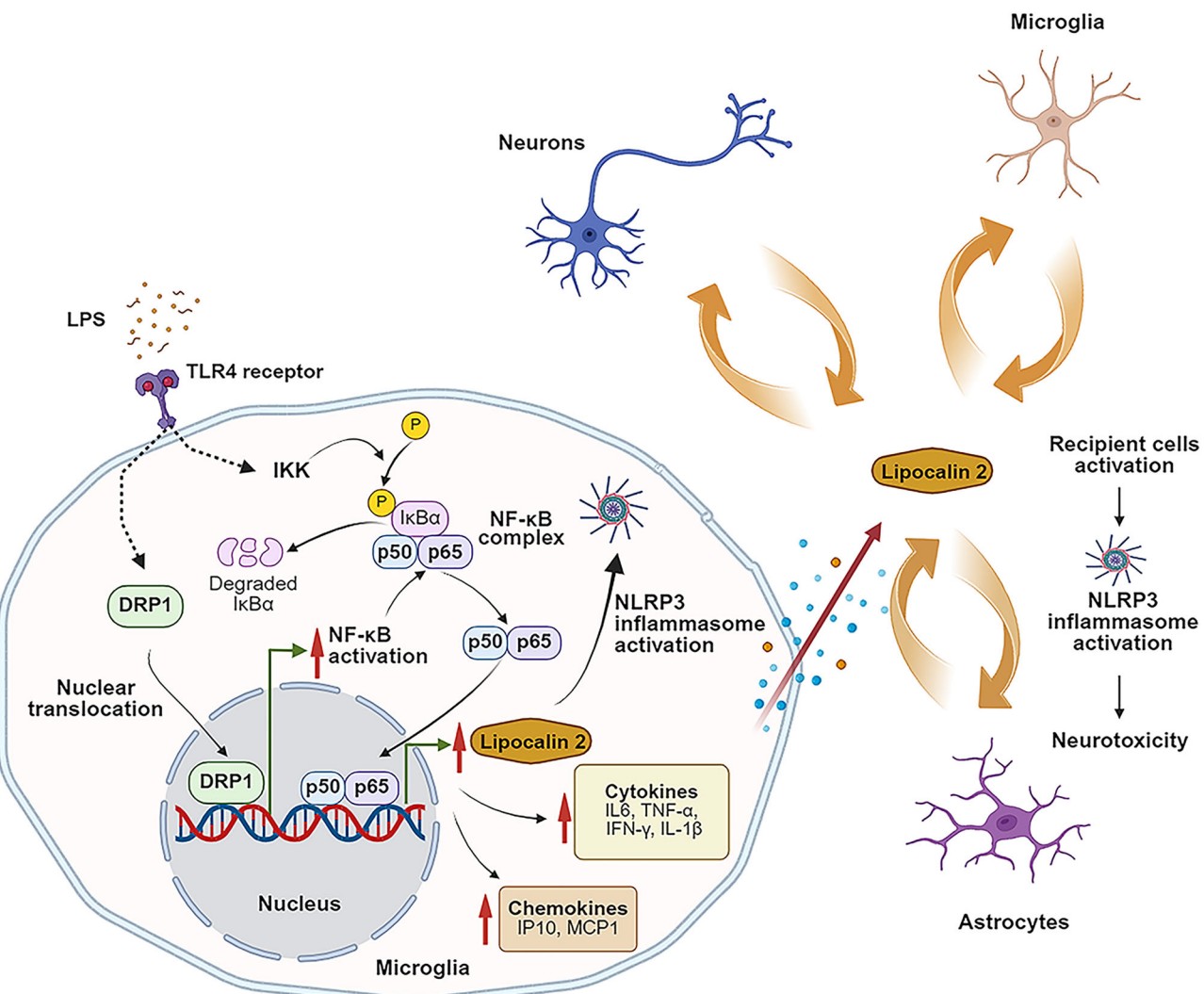

**Fig. 8 | Conceptual framework illustrating how DRP1 induces neuroinflammation via the NF-κB–lipocalin-2 axis.** Upon stimulation with proinflammatory signals such as LPS through TLR4, DRP1 translocates from the cytosol into the nucleus, where it binds to the *Rela* promoter and enhances its transcription. Increased NF-κB p65 subsequently drives the expression of downstream inflammatory cytokines that can ultimately contribute to neurotoxicity. Microglia express higher basal levels of TLR4 than astrocytes and therefore respond earlier and more robustly to LPS. In contrast, astrocytic activation requires "priming" by cytokines released from activated microglia[72–74], making microglia the predominant initiators of the inflammatory cascade. Among NF-κB–regulated genes, *Lcn2* (encoding lipocalin-2) is the most inducible in a cell-type-dependent manner. Once secreted, lipocalin-2 further amplifies neuroinflammation in neighboring cells. This graph was created in BioRender. Lai, Y. (2026) https://BioRender.com/meryu9q.

*Dnm1l*[+/−] mice (Supplementary Fig. 10). Consistent with these results, no genotypic differences between *Dnm1l*[+/−] and *Dnm1l*[+/+] littermates were detectable for the following analyses: qPCR for mitochondrial dynamics regulators (Supplementary Fig. 11), immunoblotting for phosphorylation of DRP1-616 and DRP1-637 - the two major post-translational modifications (Supplementary Fig. 12), qPCR for *Nfkbia* and *Rela* transcripts as well as immunoblotting for their respective proteins IκBα and NFκB-p65 (Supplementary Fig. 13). In combination,

these results indicate that *Dnm1l*[+/−] mice were indistinguishable from their WT littermates.

Collectively, our results establish DRP1 as a transcriptional regulator of proinflammatory signaling. NF-κB has been studied extensively since its discovery nearly four decades ago and is now firmly established as a master regulator of inflammation, controlling a broad repertoire of pro- and anti-inflammatory genes[3]. Its actions are highly context and cell-type-dependent. For example, in neurons, NF-κB is essential for synaptic plasticity and survival[4], and in PD models, NF-κB activation can promote neuroprotection, whereas inhibition can be detrimental[4]. In contrast, inhibition of *Lcn2*, a downstream NF-κB target, has consistently been shown to reduce neuroinflammation and enhance neuroprotection. Because *Lcn2* operates downstream of NF-κB and regulates a narrower set of inflammatory outputs, it may represent a more selective therapeutic target.

By demonstrating that DRP1 regulates the NF-κB–lipocalin-2 axis, a central pathway in neuroinflammation, this study reveals a previously unrecognized mechanism linking mitochondrial dynamics machinery to inflammatory gene expression. Given the central role of DRP1 in cellular stress responses, these findings suggest that DRP1-mediated transcriptional control may contribute to a wide spectrum of inflammatory and neurodegenerative diseases. Indeed, DRP1 inhibition has shown therapeutic potential across numerous models, including PD[60,61], Alzheimer's disease[30,62,63], Huntington's disease[30,59] and amyotrophic lateral sclerosis[30,64], where it protects against mitochondrial fragmentation, mitochondrial function, neuroinflammation, and neurodegeneration. The present work further expands the protective repertoire of DRP1 inhibition by establishing its transcriptional role in regulating neuroinflammatory pathways.

## Limitations of the study

In addition to some outstanding questions noted above, this study has several limitations. First, the mechanisms responsible for DRP1 nuclear translocation remain unknown. DRP1 undergoes numerous post-translational modifications, including phosphorylation, SUMOylation, ubiquitination, S-nitrosylation, and O-GlcNAcylation, making it challenging to pinpoint the specific modification(s) or combination that mediate nuclear import. It is also possible that this translocation is mediated by a mechanism that is independent of DRP1 post-translational modification. Second, our LPS paradigm models an acute early stage of neuroinflammation, it may not fully recapitulate the chronic inflammatory milieu of PD. Third, although this study focuses on the role of DRP1-NF-κB-lipocalin 2 axis in mediating neuroinflammation, as a transcriptional factor, DRP1 may regulate additional stress-responsive genes. A broader transcriptional analysis will be necessary to define the full extent of DRP1's nuclear functions.

## Methods

### Animal models and LPS treatment

All mice used in this study were bred, maintained, and characterized in the animal care facility at Florida International University (FIU). Mice were housed under a standard 12 h light/12 h dark cycle at controlled ambient temperature and humidity. All procedures were approved by, and conducted in accordance with, the FIU Institutional Animal Care and Use Committee (Protocol Approval #: IACUC-24-088-CR01).

**Dnm1l-deficient mouse models.** The generation of *Dnm1l*[+/−] mice with germline deletion of exon 2 of the *Dnm1l* gene has been described in our recent publication[24]. To generate conditional DRP1 knockout mouse lines, we contracted the European Conditional Mouse Mutagenesis Consortium (EUCOMM) to produce *Dnm1l-Loxp*[+/+] mice using "knockout-first" technology[65]. In this approach, the *tm1a* allele, containing an IRES:*lacZ* gene-trap cassette and a floxed promoter-driven neo cassette inserted into an intron of *Dnm1l*, was crossed to a FLP recombinase–expressing mouse to convert the *tm1a* allele into the conditional *tm1c*[24]. The resulting *Dnm1l-Loxp*[+/+] (tm1c) mice were crossed with either *Cx3Cr1-Cre*[ERT2+/−] mice (#021160, Jackson Laboratory) or *Aldh1L1-Cre*[ERT2+/−] mice (#031008, Jackson Laboratory) to generate *Cx3Cr1-Cre*[ERT2+/−]:*Dnm1l-Loxp*[+/−] mice ("Microglia-KO") and *Aldh1L1-Cre*[ERT2+/−]:*Dnm1l-Loxp*[+/−] ("Astrocyte-KO") mice, respectively. *Dnm1l-Loxp*[+/−] littermates were used as controls. Cre recombination was induced by intraperitoneal injection of tamoxifen (75 mg/kg, dissolved in corn oil; Sigma T2859) for five consecutive days. LPS treatment was performed three weeks after tamoxifen injection to ensure efficient *Dnm1l* knockout.

**α-Synuclein transgenic and DRP1-deficient double mutant mice.** C57BL/6N-Tg(*Thy1-SNCA*)15Mjff/J mice (#017682, Jackson Laboratory, *Dnm1l*[+/+]:*SNCA*[+/−]) were crossed with *Dnm1l*[+/−]:*SNCA*[−/−] (heterozygous DRP1-KO) mice to generate *Dnm1l*[+/−]:*SNCN*[+/−] double-heterozygous mice. *Dnm1l*[+/+]:*SNCA*[−/−] littermates served as controls.

**Lcn2-knockout mice.** B6.129P2-*Lcn2tm1Aade*/AkiJ (*Lcn2*[−/−]) mice (#024630, Jackson Laboratory) were crossed with C57BL/6 J mice to generate *Lcn2*[+/−] mice. *Lcn2*[+/−] mice were paired with *Lcn2*[+/−] mice to maintain the colony.

**LPS treatment.** Three-month-old mice received a single intraperitoneal (i.p.) injection of LPS (5 mg/kg; *E. coli* O111:B4, Sigma L4391) or saline (control). Male and female mice were randomized into treatment groups. Mice were euthanized 6 hours after injection. This regimen was selected because it induces rapid and sustained microglial activation in the substantia nigra without causing dopaminergic neurodegeneration at this early time point[32,33]. Brains were microdissected by regions, snap-frozen, or fixed for subsequent analysis.

### NanoString codeset

The reporter codeset used was the nCounter Mouse Neuroinflammation Panel (cat. no. 115000237, NanoString Technologies), which measures 757 target genes spanning core neuroinflammatory pathways and processes, along with 13 internal reference genes for data normalization.

### RNA extraction and NanoString nCounter analysis

Total RNA was extracted from the mouse ventral midbrain using TRIzol (Thermo Fisher Scientific) according to the manufacturer's instructions. Seventy-five nanograms of total RNA per sample was used for mRNA expression analysis on the NanoString nCounter SPRINT Profiler (NanoString Technologies). Target mRNAs were hybridized to reporter-capture probe pairs at 65 °C for 20 h, following the manufacturer's protocol. The hybridized probe–target complexes were subsequently aligned and immobilized in the nCounter cartridge, which was loaded into the nCounter system for imaging and data collection. Gene expression levels were exported in comma-separated value format. Raw digital counts were analyzed using nSolver v4.0 software (NanoString Technologies). Differential gene expression analysis was performed using a generalized linear model appropriate for count-based data. Statistical significance was determined using model-based p-values, and multiple testing was corrected using the Benjamini–Hochberg method.

### Quantitative RT-PCR

For reverse transcription, 1 μg of total RNA was converted to complementary DNA (cDNA) using the iScript Reverse Transcription Supermix (Bio-Rad). qPCR was performed using TaqMan Fast Advanced Master Mix (Thermo Fisher Scientific) and gene-specific TaqMan assays on a QuantStudio 6 real-time PCR system (Thermo Fisher Scientific). The following mouse Taqman assays were used: *Lcn2* (ID: Mm01324470_m1), *Nlrp3* (ID: Mm00840904_m1), *Il6* (ID: Mm00446190_m1), *Il1b* (ID: Mm00434228_m1), *Tnfα* (ID:

Mm00443258_m1), *Nos2* (ID: Mm00440502_m1), *Rela* (ID: Mm00501346_m1), and *Gapdh* (ID: Mm99999915_g1). *Gapdh* served as the internal control. Thermal cycling conditions were: 50 °C for 2 min, 95 °C for 2 min and 40 cycles of 95 °C for 1 s and 60 °C for 20 s. Relative quantification was calculated using the $2^{-\Delta\Delta CT}$ method.

### Meso Scale Discovery (MSD) multiplex immunoassay

*Dmn1l*[+/−] and *Dmn1l*[+/+] mice (3-month-old) were treated with LPS (5 mg/kg, i.p.) or saline control for 6 h before ventral midbrains were collected. Tissues were weighed and homogenized in 1x Tris lysis buffer (150 mM NaCl, 1 mM EDTA, 1 mM EGTA, 1% Triton X-100, 20 mM Tris-HCl, pH 7.5) supplemented with protease inhibitors at a ratio of 75 mg tissue per 0.5 mL ice-cold buffer. Samples were briefly sonicated, incubated on ice for 30 min, and centrifuged at 20,000 x *g* for 10 min at 4 °C. Supernatants were collected, and protein concentrations were determined using the BCA assay. For in vitro studies, primary microglia isolated from newborn pups were plated in 96-well plates and treated with 100 ng/mL LPS or vehicle for 6 h. After treatment, conditioned media containing secreted cytokines were collected and transferred to new plates. Pro-inflammatory cytokine levels in tissue lysates and conditioned media were measured using a custom U-PLEX mouse panel (lipocalin-2, IL-6, IL-1β, TNF-α, IP-10, MCP-1/CCL2, and IFN-γ) on the Meso Scale Discovery (MSD) platform following the manufacturer's instructions. Plates were read using the MSD QuickPlex SQ 120, and data were analyzed with Discovery Workbench software.

### DAB staining and Sholl analysis

Mice were perfused and fixed as previously reported[61]. Mice were perfused with 0.9% saline followed by 4% paraformaldehyde (PFA) in 0.1 M phosphate buffer. Brains were removed and post-fixed in 4% PFA overnight at 4 °C. Tissues were cryoprotected by sequential overnight incubation in 15% and 30% sucrose in 0.1 M phosphate buffer, then snap-frozen in 2-methylbutane at −55 °C. Frozen brains were embedded in Optimal Cutting Temperature (OCT) and cryosectioned at 30 μm using a CryoStar NX70 cryostat (Thermo Fisher Scientific). Coronal sections were collected free-floating in 0.1 M Tris-buffered saline (TBS).

Midbrain sections were washed in TBS and incubated in 10% methanol + 0.3% $H_2O_2$ for 10 min at room temperature to quench endogenous peroxidases. Sections were then blocked in 5% normal goat serum (NGS) and 0.1% Triton X-100 in TBS for 1 h at room temperature, followed by overnight incubation at 4 °C with primary antibody against IBA1 (1:1000, cat. no. 019-19741, FUJIFILM Wako Chemicals) prepared in 2% NGS and 0.1% Triton X-100 in TBS.

After washing with 0.1 M TBS, sections were incubated for 1 h at room temperature with biotinylated goat anti-rabbit IgG (1:200, cat. no. BA-1100, Vector Laboratories) in 2% NGS/0.1 M TBS. Sections were then washed and incubated with ABC reagent (Vectastain ABC kit, cat. no. PK-4000, Vector Laboratories) for 1 h, followed by development with 3,3′-diaminobenzidine (DAB) substrate (cat. no. SK-4100, Vector Laboratories) for 10 min.

Slides were rinsed in water and dehydrated through graded ethanol (70%, 90%, 100%). Coverslips were applied using DPX mounting medium (Sigma). Brightfield images were obtained using a Nikon microscope with a 100× objective. Sholl analysis of microglial morphology was performed using the Sholl Analysis plug-in for Fiji (http://fiji.sc/Sholl_Analysis) as previously described[66]. Thirty microglia located in the substantia nigra were analyzed per group.

### Immunofluorescence-laser microdissection (IF-LMD)

Serial coronal sections of midbrains containing the substantia nigra (SN, 10 μm) were cut on a CryoStar NX70 cryostat (Thermo Fisher Scientific). Eight sections per brain hemisphere were mounted onto PEN membrane slides (4 μm; cat. no. 11600288, Leica) for IF-LMD, as previously described[24]. Sections were fixed in cold acetone for 5 min,

rinsed in DEPC-treated 1× PBS, and blocked with 5% normal goat serum (NGS) for 30 min at room temperature. Sections were then incubated with primary antibodies for 1 h, followed by secondary antibodies for 30 min at room temperature, with DEPC-treated PBS washes between each step. After counterstaining with DAPI (10 μg/mL) for 5 min, sections were dehydrated through graded alcohols (30 s each) and air-dried. Primary antibodies used were: Chicken anti-tyrosine hydroxylase (1:200, Abcam 76442), Rabbit anti-IBA1 (1:100, Wako 019-19741), Mouse anti-GAD67 (1:50, Millipore MAB5406), and Chicken anti-GFAP (1:200, Thermo Fisher PA1-10004). Secondary antibodies (all 1:100 dilution) included: Alexa Fluor 568 goat anti-chicken (Thermo Fisher A11041), Alexa Fluor 488 goat anti-rabbit (Thermo Fisher A11008), and Alexa Fluor 488 goat anti-mouse (Thermo Fisher A11029). Immunostained dopaminergic neurons (SNpc), GABAergic neurons (SNpr), and microglia and astrocytes (substantia nigra) were isolated by laser microdissection using the Leica LMD6 system. A total of 40 cells per cell type were collected for downstream analyses.

### Reverse transcription and cDNA preamplification using Smart-seq2 method

Laser microdissected cells (40 cells per cell type) were collected into 0.2-mL thin-wall PCR tubes containing 2 μL of mild hypotonic lysis buffer (0.2% Triton X-100 and 2 U/μL RNase inhibitor; Clontech). Cell lysates were processed for reverse transcription and cDNA pre-amplification following the Smart-seq2 protocol[67,68]. For first-strand synthesis, lysates were mixed with 1 μL oligo-dT primer (10 μM; Integrated DNA Technologies) and 1 μL dNTP mix (10 mM; Thermo Fisher Scientific), denatured at 72 °C for 3 min, and immediately cooled on ice. Reverse transcription was performed using SuperScript II reverse transcriptase (Thermo Fisher Scientific) with the following program: 42 °C for 90 min, 10 cycles of 50 °C for 2 min and 42 °C for 2 min, and 70 °C for 15 min for enzyme inactivation. Subsequently, cDNA pre-amplification was carried out using KAPA HiFi HotStart ReadyMix (Roche) under the following conditions: 98 °C for 3 min, 24 cycles of 98 °C for 15 s, 67 °C for 20 s, and 72 °C for 6 min, followed by a final extension at 72 °C for 5 min. PCR products were purified using AMPure XP beads (Beckman Coulter) according to the manufacturer's instructions and eluted in 15 μL TE buffer. The resulting preamplified cDNA was used for downstream qPCR analysis as described above.

### Primary mouse microglia

Primary microglia were isolated from postnatal day 0–1 mouse pups as previously described[69]. Briefly, following decapitation, whole brains were removed, meninges were carefully dissected away, and tissues were incubated in 0.25% trypsin at 37 °C for 15 min with gentle inversion every 5 min. The digested tissue was triturated and passed through a 70-μm cell strainer before plating into culture flasks. Mixed glial cultures were maintained for 16 days in DMEM/F12 supplemented with 10% heat-inactivated fetal bovine serum (HI-FBS), 2 mM L-glutamine, 1 mM sodium pyruvate, 1% non-essential amino acids, and 1% penicillin–streptomycin. A full medium change was performed on day 6 after initial plating. On day 16, microglia were isolated from the mixed glial cultures using a CD11b immunomagnetic positive selection kit (cat. no. 18970, STEMCELL Technologies) and plated for experiments. For all treatments, HI-FBS was reduced to 2%. Microglial purity (>95%) was confirmed by IBA1 immunofluorescence.

### Adult microglia isolation and mitochondrial respiration assay

Wild-type mice (2–4 months old) were injected with LPS (5 mg/kg, i.p.) or saline. Six hours after injection, microglia were isolated using a modified gradient centrifugation protocol[70,71]. Mice were perfused with ice-cold sterile 0.9% saline, and brains were rapidly removed following decapitation and placed into ice-cold Hibernate A medium supplemented with B27, GlutaMAX, penicillin–streptomycin, and DNase I (HABG)[71]. Brains were transferred onto sterile filter paper pre-

wetted with HABG on a cooled platform and finely minced with a sterile scalpel. Minced tissue was then placed into a dissociation solution (DMEM/F12 containing 1 mg/mL papain, 1.2 U/mL dispase II, and 20 U/mL DNase I) and incubated at 30 °C for 20 min, with gentle inversion every 5 min. After digestion, a neutralization solution was added and the tissue suspension was transferred to new tubes containing 2 mL HABG.

Tissues were gently triturated with a fire-polished 9-inch Pasteur pipette, and the resulting cell suspension was collected and filtered through a 70-µm strainer. The filtrate was centrifuged at 400 x g for 5 min at 4 °C, and the pellet was resuspended in 6 mL HABG. The cell suspension was carefully layered onto a freshly prepared four-layer OptiPrep density gradient (7.4%, 9.9%, 12.4%, and 17.4% OptiPrep in Hibernate A/B27)[70].

Gradients were centrifuged at 800×g for 30 min at 22 °C, using minimal acceleration/deceleration (setting 0). The upper layers were aspirated, and the microglia-enriched lower fractions[70] were collected and diluted in 5 mL cold HABG, followed by centrifugation at $400 \times g$ for 10 min to remove residual OptiPrep. Pellets were resuspended, counted, and plated into Seahorse assay plates. Plates were centrifuged at $1000 \times g$ for 15 min to promote adhesion, then topped with assay medium and equilibrated in a non-CO₂ incubator for 30 min. Mitochondrial respiration was assessed using a standard Mito Stress Test on the Agilent XF$^e$96 Extracellular Flux Analyzer, with extended basal measurements.

### Mitochondrial respiration assay

Mitochondrial respiration was assessed using Agilent XF$^e$ 96 Extracellular Flux Analyzer (Agilent Inc) as previously reported[69]. Briefly, primary microglia isolated from *Dnm1l*$^{+/+}$ and *Dnm1l*$^{+/-}$ mouse pups were seeded into XF Cell Culture Microplates and treated the following day with LPS (100 ng/mL) for 2 h, 6 h, or 24 h. On the day of the assay, cells were washed twice with XF assay medium (DMEM supplemented with 1 mM pyruvate, 2 mM GlutaMAX, 2 mM glucose, and 2 mM HEPES, pH 7.2). Plates were then equilibrated in a non-CO₂ incubator for 30 min before being loaded into the Analyzer. A standard XF Cell Mito Stress Test was performed, with sequential injections of oligomycin, carbonyl cyanide p-trifluoromethoxyphenylhydrazone (FCCP), and rotenone/antimycin A to evaluate ATP-linked respiration, maximal respiration, and non-mitochondrial oxygen consumption, respectively.

### Immunostaining

**Primary microglia.** Primary mouse microglia were cultured on poly-D-lysine–coated borosilicate coverslips in 24-well plates. After treatment with LPS (100 ng/mL) for 2, 6, or 24 h, cells were fixed and immunostained as previously described[69]. Primary antibodies included: Rabbit anti-TOM20 (1:1000, Abcam ab186735), Mouse anti-DRP1 (1:500, BD Biosciences 611113), Rabbit anti-DRP1 (1:500, Novus NB110-55288), and Goat anti-IBA1 (1:500, Novus NB100-1028). Corresponding Alexa Fluor 488, 594, or 633 secondary antibodies (Invitrogen) were used at 1:1000. Coverslips were incubated with primary antibodies overnight at 4 °C on an orbital shaker, followed by secondary antibody incubation at room temperature for 1 h. Mounted coverslips were prepared using ProLong™ Gold Antifade Mountant with DAPI (Thermo Fisher Scientific).

**Brain tissue sections.** Mice were perfused with 0.9% saline followed by 4% paraformaldehyde (PFA) in 0.1 M phosphate buffer 6 h after LPS exposure. Brains were post-fixed, snap-frozen in 2-methylbutane, embedded in OCT, and cryosectioned at 30 µm. Free-floating coronal sections were collected in 0.1 M TBS. Sections were blocked using the Mouse-on-Mouse (M.O.M.) reagent (Vector Laboratories, BMK2202) and incubated with the following primary antibodies: Mouse anti-DRP1 (1:500, BD Biosciences 611113), and Rabbit anti-IBA1 (1:1000, FUJIFILM

Wako 019-19741). Alexa Fluor secondary antibodies (Invitrogen) were used at 1:1000. After immunostaining, sections were incubated with DAPI (5 µg/mL) for 10 min, washed, and mounted with ProLong™ Gold Antifade Mountant. All fluorescent images were captured using an Olympus Fluoview FV10i confocal microscope equipped with a 60× objective.

### Mitochondrial morphology and network analysis

Mitochondrial morphology and network structure were quantified using the Mitochondrial Network Analysis (MiNA) plugin in Fiji, as previously described[24]. Mitochondria were visualized by TOM20 immunofluorescence (Abcam ab186735) and imaged using an Olympus Fluoview confocal microscope. Z-stack images were used for all MiNA analyses. MiNA-derived parameters included: Mean rod/branch length: the average length of all mitochondrial rods and branches. Mean branches per network: the average number of branches within each mitochondrial network. Individual mitochondrial morphology was further assessed in Fiji as previously reported[69]. Two metrics were quantified: Roundness = $4 \times$ (Area) / ($\pi \times$ [Major axis]²) and Aspect ratio = Major axis / Minor axis. Values for both parameters approach 1 as mitochondria become more circular.

### Quantification using Imaris

DRP1 nuclear translocation in primary microglia and in microglia within the substantia nigra of adult mouse brains was quantified using Imaris (Oxford Instruments). Z-stack confocal images were acquired using a 60× objective lens with 3× optical zoom. Briefly, nuclear and cellular 3D surfaces were reconstructed from the blue (nuclei) and cyan (cell surface) channels using the "Surfaces" module. DRP1 puncta were detected from the red fluorescence channel using the "Spots" module, with identical threshold settings applied across all treatment groups. DRP1$^{Nuc}$ was defined as the number of DRP1 puncta located within the nuclear volume. DRP1$^{Cyto}$ was defined as the number of DRP1 puncta located within the cytoplasmic volume, calculated as total cellular puncta minus DRP1$^{Nuc}$.

### Nuclear and cytoplasmic fractionation

To quantitatively assess DRP1 nuclear enrichment following LPS exposure, nuclear and cytoplasmic extracts were prepared from LPS-treated primary mouse microglia; vehicle-treated cells served as controls. Cells were harvested into 100 µL Cytoplasmic Extract Buffer (10 mM HEPES, pH 7.6; 60 mM KCl; 1 mM EDTA; 0.075% NP-40; 1 mM DTT; 1 mM PMSF) and incubated on ice for 10 min to permeabilize the plasma membrane. Lysates were centrifuged at $200 \times g$ for 4 min at 4 °C to separate the cytoplasmic fraction (supernatant) from the intact nuclei (pellet). The supernatant was collected as crude cytoplasmic lysate. Nuclear pellets were washed once with Cytoplasmic Extract Buffer lacking NP-40, followed by centrifugation at $200 \times g$ for 4 min at 4 °C. Pellets were then resuspended in Nuclear Extract Buffer (20 mM Tris-HCl, pH 8.0; 420 mM NaCl; 1.5 mM MgCl₂; 0.2 mM EDTA; 0.5 mM DTT; 25% glycerol; 1 mM PMSF). Nuclear suspensions were homogenized on ice using a Qsonica sonicator for 3 cycles of 30 s ON/30 s OFF at 60% amplitude. Both crude cytoplasmic and nuclear suspensions were centrifuged at $21,000 \times g$ for 30 min at 4 °C to remove debris. The resulting supernatants, cytoplasmic extracts, and nuclear extracts were aliquoted and stored at −80°C for subsequent analyses.

### Capillary gel-free immunoblotting

Capillary immunoblotting was performed using the Jess™ system (ProteinSimple) with the 12–230 kDa Fluorescence Separation Module (cat. no. SM-FL004), as previously described[24]. Protein concentrations were quantified using the Micro BCA kit (cat. no. 23235, Thermo Fisher Scientific). Cytoplasmic extracts, nuclear extracts, or whole-cell lysates (prepared in RIPA buffer) were diluted in 0.1× sample buffer and mixed with 5× Fluorescent Master Mix (cat. no. PS-ST01EZ, ProteinSimple) to

final concentrations of 0.5 μg/μL and 1.2 μg/μL, respectively. Samples were boiled at 95 °C for 5 min, and 3 μL of each sample were loaded into the capillary system.

Primary antibodies included: Rabbit anti-DRP1 (1:100, Novus NB110-55288), Rabbit anti-NF-κB p65 (1:25, CST 8242), Rabbit anti-Histone H3 (1:100, Abcam ab1791), and Rabbit anti-GAPDH (1:100, CST 2118). Anti-rabbit secondary antibodies were supplied through the appropriate Detection Module (cat. no. DM001, ProteinSimple). The Protein Normalization Module (cat. no. DM-PN02, ProteinSimple) was used for quantitative signal normalization. Data acquisition and analysis were carried out using Compass software v6.0.

## SDS-PAGE immunoblotting
Mouse ventral midbrains were resuspended in RIPA lysis buffer and sonicated on ice using a Qsonica sonicator for 5 cycles of 10 s ON/10 s OFF at 50% amplitude. Lysates were rotated at 4 °C for 30 min, followed by centrifugation at 13,000 × g for 15 min at 4 °C. Primary microglia were lysed directly in RIPA buffer for 5 min on ice before centrifugation. Protein concentrations for all samples were determined using the Pierce™ BCA Protein Assay (Thermo Fisher Scientific). Proteins were resolved by 12% SDS-PAGE or 15% SDS-PAGE (for p62) and transferred onto 0.45-μm PVDF membranes (Immobilon-P, MilliporeSigma). Total protein levels were assessed using Revert™ 700 Total Protein Stain (LI-COR Biosciences) according to the manufacturer's instructions. Membranes were blocked for 1 h at room temperature in Intercept® Blocking Buffer (LI-COR Biosciences) and incubated overnight at 4 °C with the following primary antibodies: Rabbit anti-pDRP1-Ser616 (1:1000, CST 4494), Rabbit anti-pDRP1-Ser637 (1:1000, GeneTex GTX01567), Rabbit anti-NF-κB p65 (1:1000, CST 8242), Mouse anti-IκBα (1:1000, Novus NB100-56507), and Rabbit anti-p62 (1:1000, MBL PM045). After washing with TBS-T, membranes were incubated for 1 h at room temperature with the appropriate secondary antibodies: Anti-rabbit IgG (H + L) DyLight® 800 (1:10,000; or 1:5000 for p62; CST 5151), Anti-mouse IgG (H + L) DyLight® 800 (1:10,000; CST 5257). Blots were imaged using the LI-COR Odyssey Imaging System, and band intensities were quantified using Image Studio™ Software v6.0 (LI-COR Biosciences).

## Chromatin immunoprecipitation (ChIP) assay
A total of $6 × 10^6$ primary microglia were treated with LPS (100 ng/mL) for 6 h; cells treated with medium alone served as controls. After treatment, cells were washed twice with PBS containing protease inhibitors and processed for the ChIP assay. For crosslinking, cultures were incubated in medium containing 1% (w/v) formaldehyde (Sigma-Aldrich) at 37 °C for 15 min (NF-κB p65) or 30 min (DRP1). Crosslinking was quenched with 125 mM glycine for 5 min with shaking. Cells were collected by scraping, centrifuged, washed twice with cold PBS plus protease inhibitors (Pierce, Thermo Fisher Scientific), and resuspended in cell lysis buffer containing 1% SDS, 10 mM EDTA, 50 mM Tris-HCl (pH 8.0) and protease inhibitors. Lysates were incubated on ice for 10 min. Chromatin shearing was performed using a Qsonica sonicator at 4 °C for 30 cycles of 10 s ON/30 s OFF at 60% amplitude, resulting in cell lysis and DNA fragmentation. Lysates were centrifuged at 18,000 × g for 10 min at 4 °C, and the clarified supernatant was collected. The supernatant was diluted 10-fold with ice-cold ChIP dilution buffer (1% Triton X-100, 1.2 mM EDTA, 167 mM NaCl, 16.7 mM Tris-HCl, pH 8.0, with protease inhibitors). A 100-μL aliquot was set aside as the Input.

The remaining supernatant was pre-cleared by incubation with sheared salmon sperm DNA-coated protein A agarose beads (Pierce, Thermo Fisher Scientific) for 2 h at 4 °C with rotation. The pre-cleared chromatin was then divided into equal aliquots and incubated overnight at 4 °C with: IgG control (2 μg; CST #2729), Anti-NF-κB p65 antibody (2 μg; CST #8242), and Anti-DRP1 antibody (2 μg; CST #8570). Immune complexes were captured using sheared salmon sperm

DNA-coated protein A agarose beads for 2 h at 4 °C with rotation. Bead-bound complexes were pelleted at 500 × g for 2 min and washed sequentially: twice with low-salt wash buffer (150 mM NaCl), twice with high-salt wash buffer (500 mM NaCl), and twice with TE buffer. Chromatin was eluted from the beads using freshly prepared elution buffer (1% SDS, 0.1 M NaHCO₃). Crosslinks were reversed by incubation with 0.2 M NaCl at 65 °C overnight. Samples were then treated with proteinase K at 45 °C for 2 h, followed by phenol–chloroform extraction and ethanol precipitation. Purified DNA was resuspended in TE buffer and used for quantitative PCR analysis.

## Quantitative real-time PCR for ChIP assay
Quantitative real-time PCR was performed using KAPA SYBR FAST qPCR Master Mix (Sigma) in a 10-μL reaction containing purified ChIP DNA (NF-κB p65–bound, DRP1-bound, or IgG control) and gene-specific primers (Supplementary Data 2). Reactions were run on a QuantStudio 6 real-time PCR system (Thermo Fisher Scientific) using the following cycling program: 95°C for 3 min (1 cycle), 40 cycles of 95°C for 3 s, Primer-specific-annealing temperature Ta (Supplementary Data 2) for 20 s, 72°C for 1 s. Ct values were obtained using QuantStudio Software v1.3. Data analysis:

Fold enrichment was calculated by normalizing ChIP Ct values to the Input DNA. The normalized ΔCt for each ChIP reaction was computed as:

$$\Delta Ct_{\text{normalized ChIP}} = Ct_{\text{ChIP}} - \left( Ct_{\text{Input}} - \log_2(\text{Input dilution factor}) \right).$$ The

Input dilution factor is defined as the volume used for each IP/the volume of Input saved for further analysis. Volume saved = 100 μL and volume used for each IP = 900 μL. Thus, the Input dilution factor was 9, and the equation was $\Delta Ct_{\text{normalized ChIP}} = Ct_{\text{ChIP}} - \left( Ct_{\text{Input}} - \log_2(9) \right)$. The

percentage of Input was calculated as: Input % = $2^{-\Delta Ct_{\text{normalized ChIP}}} × 100$. The Input % value reflects the enrichment of NF-κB p65 or DRP1 (relative to IgG control) at the promoter region of their target gene(s). Higher Input % values indicate greater transcription factor or DRP1 binding.

## Construction of mouse *Rela* promoter-driven reporter construct
Genomic DNA isolated from *Dnm1l*⁺/⁺ (WT) mice was used as the template for PCR amplification of the DRP1-binding region within the mouse *Rela* promoter. The primer sequences were as follows: sense, 5′-ggtaccACC TGC GGA GCT TGT AG −3′, and antisense, 5′- aagcttGCT AAA GTA AAG CCA TTC G −3′. The sense primer contained a *KpnI* restriction site, and the antisense primer contained a *HindIII* site. PCR was performed using Platinum II Taq Hot-Start DNA Polymerase (Invitrogen, cat. no. 14000012) according to the manufacturer's instructions. Amplified fragments were first ligated into the PCR2.1 vector (Invitrogen, cat. no. K203001). Following digestion with the corresponding restriction enzymes and gel purification (Qiagen, cat. no. 28704), the inserts were subcloned into the pGL4.20[*luc2*/Puro] vector (Promega, cat. no. E6751). The DNA sequence of the final construct was confirmed by whole-plasmid sequencing (Eurofins Genomics).

## Cell transfection and luciferase assay
HT22 mouse hippocampal cells were plated at a density of $1 × 10^4$ cells per well in 96-well plates. After 24 h, cells were transfected with 0.1 μg of either the pGL4.20[*luc2*/Puro] control vector or the pGL4.20[*luc2*/Puro]-*Rela* construct in the presence or absence of *Dnm1l*-specific siRNA (50 nM) or a scrambled siRNA control using jetPRIME (SATOR-IUS, cat. no. 10100001), following the manufacturer's protocol. The siRNA reagents included SMARTpool siGENOME mouse *Dnm1l* siRNA (Dharmacon/Horizon, cat. no. M-054815-01), which is a mixture of four siRNA targeting a single gene to enhance efficiency and specificity of *Dnm1l*-knockdown. The siGENOME Non-Targeting siRNA Control Pool (cat. no. D-001206), the scramble control, has a minimum of four

siRNAs designed to target no known genes in human, mouse or rat genes. Firefly luciferase activity was measured 48 h post-transfection using the ONE-Glo™ EX Luciferase Assay System (Promega, cat. no. E8110), following the manufacturer's instructions. Knockdown efficiency of *Dnm1l* was confirmed by qRT-PCR analysis of *Dnm1l* mRNA levels.

## Statistics

All values are presented as mean ± SEM. Differences between group means were analyzed using t-tests or one-, two-, or three-way ANOVA, with different mouse genotypes, treatment, and or time as independent factors. When ANOVA indicated a significant main effect or interaction, Tukey's post hoc test was used for pairwise comparisons. All datasets were evaluated for normality and homogeneity of variance. When these criteria were not met, nonparametric tests were applied, specifically the Kruskal–Wallis ANOVA followed by Dunn's post hoc test. In all analyses, the null hypothesis ($H_0$) or the alternative hypothesis ($H_1$) was accepted with an α-error (false-positive) ≤ 5% and a β-error (false-negative) ≤ 20%, respectively. F-values, degrees of freedom, and associated p-values for all analyses are reported in Supplementary Data 3. Volcano plots were generated using the Positron integrated development environment (version 2025.11.0, build 234; Posit PBC) with Bioconductor version 3.22 and the EnhancedVolcano package version 1.27.

## Reporting summary

Further information on research design is available in the Nature Portfolio Reporting Summary linked to this article.

## Data availability

All data supporting the findings of this study are available within the article as well as Supplementary Information and Source Data files. Source data are provided with this paper.

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

## Acknowledgements

This study was supported in part by the National Institute of Environmental Health Sciences (NIEHS) of the National Institutes of Health (NIH) under Award Number R35ES030523 (to KT). The content is solely the responsibility of the authors and does not necessarily represent the official views of the NIH.

## Author contributions

K.T., Y.L., and R.F. contributed to the study conception and design. Experiments and data analysis were performed by Y.L., R.F., H.B., S.S., and E.T. Y.L. and K.T. were involved in drafting and revision of the

manuscript. All the other authors commented on previous versions of the manuscript and approved of the final version.

## Competing interests

The authors declare no competing interests.
