## [Transparent Peer Review file · Nature Communications]

DRP1 induces neuroinflammation via transcriptional regulation of NF- κ B

Corresponding Author: Professor Kim Tieu

Version 0:

Reviewer comments:

Reviewer #1

(Remarks to the Author)

Drp1 Induces Neuroinflammation via Transcriptional Regulation of NF κ B

This manuscript by Lai al. presents compelling evidence for a non-canonical role of Drp1 regulating neuroinflammation. Using transgenic mouse models of Drp1, the authors first identify lipocalin-2 as a key proinflammatory target of Drp1 in glial cells of the ventral midbrain regions following LPS stimulation. They further demonstrate that Drp1 is upregulated and rapidly translocated to the nucleus under these conditions. Notably, authors uncover that Drp1 directly binds to Rel promoter, thereby driving lipocalin-2 mediated neuroinflammatory responses. The study employs multiple methodologies and models, and the experimental design appears rigorous and well-executed. This work provides novel insights into previously unreported function of Drp-1, suggesting its dual role in mitochondrial quality control and dampening neuroinflammation. These results have important implications for development of Drp1 targeted disease modifying therapies for Parkinson's and other related neurodegenerative diseases.

Some questions and clarifications that should be addressed.

1. Authors used a 5 mg/kg LPS injection for 6 hours to evoke neuroinflammation in the brain. While the rationale for this dose is briefly discussed in context of disentangling Drp1's dual function in the Discussion section, a clear justification should also be provided in the in methods section.
2. In fig 3b, authors show LPS treatment increases Lcn2 mRNA level in microglia, astrocytes and dopaminergic neurons. The underlying mechanism driving this upregulation warrants further discussion. Specifically, is this effect mediated by neuronal Drp1 upregulation and translocation?
3. Drp1 is known to undergo multiple post-translational modifications. Does LPS-induced Drp1 translocation to nucleus require any post-translational modifications?
4. A discussion of the study's limitations and potential future directions would be valuable. This could include technical constraints, unanswered mechanistic questions, or border implications for therapeutic strategies for chronic neurodegenerative diseases.

Reviewer #2

(Remarks to the Author)

In this paper, Lai et al. presented relevant evidence indicating that DRP1, a protein regulator of mitochondrial fission, controls neuroinflammatory response by activating NF- κ B. Apparently, this mechanism is produced by the protein lipocalin-2 in microglia and astrocytes. These interesting studies propose a novel action of DRP1 on neuroinflammation and link mitochondrial function with inflammatory response in the brain. However, several points require clarification before any

publication process.

Mayor comments

-The authors declare: "Emerging evidence indicates the central role of neuroinflammation in neurodegenerative diseases such as Alzheimer's disease (AD), Parkinson's disease (PD)²". However, this is not emerging evidence, which has been observed and proposed for quite a while. The authors must clarify this information and reduce the enthusiasm of these statements.

-The authors must discuss the antioxidant factor NRF2's contribution to NF-κB regulation and its involvement in the mechanism explored in this manuscript. Several studies suggest that NRF-2 modulates the NF-κB response, decreasing inflammatory signaling. If we consider the scope of this paper, perhaps NRF2 could be playing a role, or at least this action should be experimentally discarded.

-Did the authors test Lcn2 levels in any AD mouse models? If not, it is better that the paper focus on PD effects related to Lcn2 and not AD.

-The authors need to include specific evidence regarding the neurodegenerative role of lipocalin 2 in PD.

-As the authors have investigated DRP1 involvement in neuroinflammatory action, they should be testing the effects on mitochondrial dynamics. They need to show if the change in DRP1 affects the expression of the other mitochondrial dynamic regulators (Opa1, Mfn1/2, Fis1).

-Also, the authors should specify how DRP1 activation induced by several phosphorylation statuses could be involved in the neuroinflammatory regulation induced by NF-κB activation. At least two specific phosphorylation residues (616 or 636) can be affected by oxidative stress or other stressors important in neuroinflammation.

-The analysis of mitochondrial dynamics presented in Fig.S3 must be augmented using the macro tool for Fiji-ImageJ MiNA analysis.

Reviewer #3

(Remarks to the Author)

Lai and colleagues highlight a novel role for dynamin-related protein 1 (DRP1), previously known as a key mediator of mitochondrial fission and autophagy flux, now proposed to act as a transcription factor regulating the expression of the Lipocalin 2 gene through the control of Rela (encoding NF-κB). The authors employ a comprehensive set of state-of-the-art techniques, including both constitutive and inducible transgenic mouse models, digital PCR-based quantification of proinflammatory mRNA markers in ventral midbrain samples and laser-microdissected brain cells, high-resolution IMARIS confocal imaging, and capillary immunoblotting employing samples from primary microglial cultures. Although the study supports the identification of a novel function for DRP1 in neuroinflammation, there are several concerns that should be addressed prior to publication.

General Comments:

1. Although the topic is briefly addressed in the Discussion, the manuscript would benefit from a more clearly defined conceptual framework focused on general neuroinflammation in which microglial cells are the driving force. Notably, acute intraperitoneal LPS administration does not elicit a brain-specific proinflammatory response and astrocytes findings are not explored in depth. The extrapolation of the findings to other neurodegenerative conditions such as Alzheimer's disease, Parkinson's disease, or aging appears premature and should be reconsidered. This is particularly relevant given that the study relies on a single, SNCA model of PD (Dnm11^{+/-}:SNCA^{+/-}, that is not characterized in the study) and includes only one experiment related to aging. The impact on dopaminergic neuronal survival in response to LPS is not addressed at all, therefore the relevance in PD models is not analyzed. Limiting the scope of the discussion to the specific experimental context would strengthen the manuscript's coherence and rigor.

2. Please clarify the rationale for using partial instead of full knockout Dnm11 models. In this regard, a deeper characterization of the employed mutant mice with heterozygous Dnm11 knockout is needed. From the molecular perspective, it would be important to analyze Dnm11 expression levels, both at the mRNA and protein level in ventral midbrain extracts, as well as the phosphorylation status of DRP1. Additionally, the expression of key components of the NFκB pathway (such as IKK activation, and the protein levels of IκBα, p50, and p65) should be assessed in comparison to wild-type controls. Related to mitochondrial physiology, it would be crucial to describe the impact of deleting one copy of Dnm11 gene on mitochondrial activity in the brain.

Specific comments.

Figure 1. To better illustrate the protective role of Dnm11, please provide a more comprehensive description of all dysregulated transcripts comparing LPS-Dnm11 ^{+/-} vs LPS-Dnm11 ^{+/+}, not just the top ten or the table. Include those validated at the protein level (e.g., IL-6, IL-1β, TNFα) in next figures.

Figure S3. To rule out mitochondrial misbalance is crucial to include positive control (e.g., rotenone or paraquat) to validate OCR respiration measures. Also, autophagy should be analyzed, especially considering previous work by the same group showing that partial Dnm11 knockout improves autophagic flux independently of mitochondrial function.

Figure 6. Regarding panel 6e, why are DRP1 nuclear levels similar between Dnm11^{+/-} and saline-treated controls? There is a reduction only in the cytosolic fraction. This analysis should be repeated in whole-cell extracts to assess total protein levels (this is closely related to the second general comment). Figure S4. An explanation addressing why Histone H3 levels are reduced in nuclear extracts from mutant mice compared to wild-type controls could be added.

Figures 6 and 7. Since LPS can activate NF κ B independently of DRP1, it is crucial to show that the identified DRP1 motif is functional. Therefore, the proposed DRP1 nuclear translocation and binding to RelA regulation sites could be strengthened by: Showing p65 protein levels in both cytosol and nucleus (Figure 6e–f), highlighting the sequence of the DRP1 motif identified by P7 primers (Figure 7), demonstrating that DRP1 binding leads to increased RelA transcription, to support the claim made in line 327.

Minor Points:

Line 38: “Low-grade chronic inflammation” better describes the inflammatory state in neurodegenerative diseases.

Line 65: Please elaborate on the neurotoxic role of Lipocalin-2.

Line 71: Use “levels are” instead of “levels is”.

Line 119: Replace “upregulation” with “was upregulated”.

Line 117: Clarify whether the global DRP1-KO mouse refers to a previous study (add reference) or unpublished data.

Line 195: The sentence appears to be cut off (please revise).

Line 432: The sentence is split across two lines (please correct).

LPS (cat# L4391, Sigma-Aldrich): Include the strain type to improve reproducibility.

Supplementary Table 1: The PDF formatting is unclear (please improve the table layout for readability maybe in an Excel file if possible).

Figure S1a: Add a scale bar to indicate magnification.

Reviewer #4

(Remarks to the Author)

Lai reports a novel mechanism underlying DRP1-mediated inflammation induced by LPS. Reduced DRP1 expression significantly suppresses LPS-induced inflammation, including the proinflammatory molecule lipocalin 2. Upon LPS stimulation, DRP1 translocates from the cytosol to the nucleus, where it binds to the promoter region of RelA (encoding NF- κ B), activating its gene products and downstream inflammatory cytokines. These findings highlight the important role of proinflammatory lipocalin 2 in the brain.

Together, this study reveals a previously unrecognized function of DRP1 in mediating neuroinflammation via the NF- κ B–lipocalin 2 axis. The experiments are well-designed and clearly presented. Only minor revisions are suggested.

Suggestions:

- Please expand the discussion on the significance and rationale for using the LPS-induced model, particularly in relation to DRP1-mediated inflammation. How well does this model reflect the in vivo mechanisms of neuroinflammation?
- It would also strengthen the manuscript to discuss the outcomes of additional models (if available) that explore the relationship between DRP1-mediated mitochondrial morphology changes, neuroinflammation, and neurodegeneration. This could provide further support for the role of DRP1 in disease progression and therapeutic targeting.

Version 1:

Reviewer comments:

Reviewer #1

(Remarks to the Author)

Authors sufficiently addressed my concerns.

Reviewer #2

(Remarks to the Author)

The authors properly reviewed all comments and included additional data to support their conclusions

Reviewer #3

(Remarks to the Author)

NCOMMS-25-49663-T

The authors have responded in detail to all the points raised, providing meaningful and relevant experiments that substantially strengthen the manuscript. In light of these revisions and the improved quality of the study, acceptance of the manuscript would now be appropriate.

Reviewer #4

(Remarks to the Author)

I do not have any further concerns. Authors responded very well to my concerns.

Response to Reviewers

Re: Manuscript resubmission (NCOMMS-25-49663-T)

We would like to express our sincere gratitude to the reviewers for their constructive, insightful, and supportive feedback, which has greatly helped us improve the manuscript. Our detailed point-by-point responses to the reviewers' comments are provided below. For clarity and convenience, new data have been included in this response letter and referenced where they have been incorporated into the revised manuscript. Text revisions in the manuscript are highlighted in red.

Reviewer #1

This manuscript by Lai al. presents compelling evidence for a non-canonical role of Drp1 regulating neuroinflammation. Using transgenic mouse models of Drp1, the authors first identify lipocalin-2 as a key proinflammatory target of Drp1 in glial cells of the ventral midbrain regions following LPS stimulation. They further demonstrate that Drp1 is upregulated and rapidly translocated to the nucleus under these conditions. Notably, authors uncover that Drp1 directly binds to Rel promoter, thereby driving lipocalin-2 mediated neuroinflammatory responses. The study employs multiple methodologies and models, and the experimental design appears rigorous and well-executed. This work provides novel insights into previously unreported function of Drp-1, suggesting its dual role in mitochondrial quality control and dampening neuroinflammation. These results have important implications for development of Drp1 targeted disease modifying therapies for Parkinson's and other related neurodegenerative diseases.

Some questions and clarifications that should be addressed.

1. Authors used a 5 mg/kg LPS injection for 6 hours to evoke neuroinflammation in the brain. While the rationale for this dose is briefly discussed in context of disentangling Drp1's dual function in the Discussion section, a clear justification should also be provided in the in methods section.

Response: The *in vivo* LPS regimen used in this study was selected partly based on previous studies, which show rapid and sustained microglia activation in the substantia nigra. Moreover, at this treatment time-point, LPS does not cause dopaminergic neurodegeneration. Prior studies using this regimen show that microglial activation emerges within hours of peripheral LPS injection, but dopaminergic neurodegeneration in the substantia nigra develops much later, approximately seven months to observe modest loss (~23%) and ten months to detect more pronounced degeneration (39–47%). Thus, this model reflects an early, pre-degenerative stage of neuroinflammation. We believe this model represents an early stage of neuroinflammation preceding cell death in PD. This is critical to detect early translocation of DRP1 to the nucleus to initiate neuroinflammation. Additionally, we found that this paradigm markedly increased pro-inflammatory cytokines in the VMB, yet mitochondrial respiration in microglia remained intact. This justification and relevant references have been elaborated more in Discussion (page 16-17) and added to the Methods section (page 23) as suggested.

2. In fig 3b, authors show LPS treatment increases *Lcn2* mRNA level in microglia, astrocytes and dopaminergic neurons. The underlying mechanism driving this upregulation warrants further discussion. Specifically, is this effect mediated by neuronal *Drp1* upregulation and translocation?

Response: Our results show that *Lcn2* expression increased in microglia, astrocytes, and DA neurons, but not GABA neurons, following LPS exposure. The lack of induction in GABA neurons suggests cell-type-specific differences in *Lcn2* inducibility, which warrants future studies. Combined with the demonstration that LPS induced DRP1 translocation in microglia, we believe that the upregulation of *Lcn2* observed in cell type is unlikely to originate from neuronal sources alone, and the elevated *Lcn2* detected *in vivo* likely arises predominantly from glial sources. This discussion has been added to page 23.

3. *Drp1* is known to undergo multiple post-translational modifications. Does LPS-induced *Drp1* translocation to nucleus require any post-translational modifications?

Response: This is an excellent question. At the moment, no known DRP1 post-translational modification has been reported to drive nuclear translocation. As mentioned by the reviewer, DRP1 is extensively regulated by multiple post-translational modifications, including phosphorylation, SUMOylation, Ubiquitination, S-nitrosylation, O-GlcNAcylation - making it challenging to pinpoint the specific modification(s) or combination that drive nuclear import. It is also possible that this translocation is mediated by a mechanism that is independent of DRP1 post-translational modification. Although this is a topic that we fully intend to pursue, it is beyond the scope of the present study. This discussion has been included in the "Limitations of the study" on page 23.

4. A discussion of the study's limitations and potential future directions would be valuable. This could include technical constraints, unanswered mechanistic questions, or border implications for therapeutic strategies for chronic neurodegenerative diseases.

Response: The following discussion has been added to page 23: "In addition to some outstanding questions noted elsewhere in this manuscript, this study has several limitations. First, the mechanisms responsible for DRP1 nuclear translocation remain unknown. DRP1 undergoes numerous post-translational modifications, including phosphorylation, SUMOylation, ubiquitination, S-nitrosylation, and O-GlcNAcylation, making it challenging to pinpoint the specific modification(s) or combination that drive nuclear import. It is also possible that this translocation is mediated by a mechanism that is independent of DRP1 post-translational modification. Second, our LPS paradigm models represent acute early stage of neuroinflammation, it may not fully recapitulate the chronic inflammatory milieu of PD. Third, although this study focuses on the role of DRP1-NF- κ B-lipocalin 2 axis in mediating neuroinflammation, as a transcriptional factor, DRP1 may regulate additional stress-responsive genes. A broader transcriptional analysis will be necessary to define the full extent of DRP1's nuclear functions."

Reviewer #2

In this paper, Lai et al. presented relevant evidence indicating that DRP1, a protein regulator of mitochondrial fission, controls neuroinflammatory response by activating NF- κ B. Apparently, this mechanism is produced by the protein lipocalin-2 in microglia and astrocytes. These interesting studies propose a novel action of DRP1 on neuroinflammation and link mitochondrial function with

inflammatory response in the brain. However, several points require clarification before any publication process.

Major comments

-The authors declare: "Emerging evidence indicates the central role of neuroinflammation in neurodegenerative diseases such as Alzheimer's disease (AD), Parkinson's disease (PD)2". However, this is not emerging evidence, which has been observed and proposed for quite a while. The authors must clarify this information and reduce the enthusiasm of these statements.

Response: Thank you for pointing this out. This sentence on page 3 has been modified to "The central role of chronic neuroinflammation in neurodegenerative diseases such as Alzheimer's disease (AD), Parkinson's disease (PD) has been well-documented."

-The authors must discuss the antioxidant factor NRF2's contribution to NF-κB regulation and its involvement in the mechanism explored in this manuscript. Several studies suggest that NRF-2 modulates the NF-κB response, decreasing inflammatory signaling. If we consider the scope of this paper, perhaps NRF2 could be playing a role, or at least this action should be experimentally discarded.

Response: To determine whether NRF2 influences NF-κB activity in our model, we analyzed our NanoString dataset and found that *Nfe2l2* (the gene encoding NRF2) mRNA levels in ventral midbrains were not significantly changed following LPS treatment as shown in the figure below (Supplementary Fig. 2a). We further validated this observation by qPCR, which similarly showed no alteration in *Nfe2l2* expression (Supplementary Fig. 2b). Together, these results indicate that NRF2 is unlikely to contribute to the neuroinflammatory response observed in this study. This discussion has been added to page 7, and the figure below has been included as **Supplementary Fig. 2**.

Supplementary Fig. 2. *Nfe2l2* levels in LPS-treated mice. Three-month-old *Dnm1*^{+/+} and *Dnm1*^{+/-} littermates LPS were injected (5 mg/kg, i.p.) or saline control, and the VMB tissue was collected 6h later for NanoString nCounter gene expression analysis (a) and qPCR validation (b).
a, *Nfe2l2* levels were quantified and normalized to the reference genes *Aars*, *Ccdc127*, *Cnot10*, *Tada2b* and *Xpnp1*. N=5-8 (2-4F & 3-4M) mice per group, two-way ANOVA.
b, *Nfe2l2* levels, normalized to *Gapdh*. N=5-7 (2-4F & 3-4M) mice per group, two-way ANOVA followed by Tukey post-hoc test.

-Did the authors test *Lcn2* levels in any AD mouse models? If not, it is better that the paper focus on PD effects related to *Lcn2* and not AD.

Response: We did not assess the levels of *Lcn2* in AD mouse models, but did cite those studies from other investigators to support the role of *Lcn2* in AD patients and experimental models to illustrate that the role of *Lcn2* is not restricted to PD. But as requested, those references and discussions related to AD have been removed throughout the manuscript.

-The authors need to include specific evidence regarding the neurodegenerative role of lipocalin 2 in PD.

Response: Lipocalin-2 is elevated in the substantia nigra of patients with PD and in MPTP-treated mice. Furthermore, *Lcn2*-KO mice protect against MPTP-induced neurodegeneration, whereas recombinant lipocalin-2 injection into the substantia nigra induces marked dopaminergic neurodegeneration. In clinical studies, cerebrospinal fluid levels of lipocalin-2 in PD patients positively correlate with α -synuclein levels and have been proposed as a potential inflammatory biomarker for PD. However, further investigation is required to clarify the mechanistic relationship between lipocalin-2 and α -synuclein-mediated neurotoxicity. In our other ongoing study, the effects of lipocalin 2 on α -synuclein pathology were investigated in the N27 rat dopaminergic neuronal cells with stable ecdysone-inducible expression of human wild-type α -syn. These cells accumulate proteinase-K resistant α -syn after two days of induction by ponasterone A, an ecdysone analogue. However, lipocalin 2 drastically increased the aggregate number, indicating the neurotoxic effects of lipocalin 2 in the presence of α -syn. This more detailed discussion has now been included in the manuscript on page 16.

-As the authors have investigated *DRP1* involvement in neuroinflammatory action, they should be testing the effects on mitochondrial dynamics. They need to show if the change in *DRP1* affects the expression of the other mitochondrial dynamic regulators (*Opa1*, *Mfn1/2*, *Fis1*).

Response: As suggested, we performed qRT-PCR to assess whether reduced DRP1 expression in *Dnm1*^{-/-} mice would alter the transcription of other key mitochondrial dynamics regulators, including *Opa1*, *Mfn2*, *Mff*, *Mief1*, *Mief2*, and *Fis1*. Our results (**Supplementary Fig. 11**) showed no significant differences in the expression of any of these genes when comparing *Dnm1*^{-/-} mice with their *Dnm1*^{+/+} littermates. These results have been added to Supplementary Information.

-Also, the authors should specify how DRP1 activation induced by several phosphorylation statuses could be involved in the neuroinflammatory regulation induced by NF-κB activation. At least two specific phosphorylation residues (616 or 636) can be affected by oxidative stress or other stressors important in neuroinflammation. DRP1 PTMs affected by oxidative stress

Response: As noted in our response to Reviewer #1, DRP1 activity can be regulated by multiple PTMs, including phosphorylation. In relation to the specific phosphorylation sites highlighted by this reviewer, we had previously examined whether phosphorylation at S616 and S637, two well-characterized PTMs known to be influenced by oxidative stress, might contribute to the results in this study. Using primary microglia treated with LPS or vehicle control, we isolated nuclear fractions and performed immunoblotting to assess levels of DRP1-pS616 and DRP1-pS637. Our results indicate that neither modification is involved in driving DRP1 translocation to the nucleus. We fully intend to further investigate the mechanisms underlying DRP1 nuclear import, which may or may not involve PTMs.

-The analysis of mitochondrial dynamics presented in Fig.S3 must be augmented using the macro tool for Fiji-ImageJ MiNA analysis.

Response: Please note that we did use this method of mitochondrial analysis as indicated in the Figure legend and the Methods section.

Reviewer #3

Lai and colleagues highlight a novel role for dynamin-related protein 1 (DRP1), previously known as a key mediator of mitochondrial fission and autophagy flux, now proposed to act as a transcription factor regulating the expression of the Lipocalin 2 gene through the control of RelA (encoding NF-κB). The authors employ a comprehensive set of state-of-the-art techniques, including both constitutive and inducible transgenic mouse models, digital PCR-based quantification of proinflammatory mRNA markers in ventral midbrain samples and laser-microdissected brain cells, high-resolution IMARIS confocal imaging, and capillary immunoblotting employing samples from primary microglial cultures.

Although the study supports the identification of a novel function for DRP1 in neuroinflammation, there are several concerns that should be addressed prior to publication.

General Comments:

1. Although the topic is briefly addressed in the Discussion, the manuscript would benefit from a more clearly defined conceptual framework focused on general neuroinflammation in which microglial cells are the driving force. Notably, acute intraperitoneal LPS administration does not elicit a brain-specific proinflammatory response and astrocytes findings are not explored in depth. The extrapolation of the findings to other neurodegenerative conditions such as Alzheimer's disease, Parkinson's disease, or aging appears premature and should be reconsidered. This is particularly relevant given that the study relies on a single, SNCA model of PD (Dnm1I+/-:SNCA+/-, that is not characterized in the study) and includes only one experiment related to aging. The impact on dopaminergic neuronal survival in response to LPS is not addressed at all, therefore the relevance in PD models is not analyzed. Limiting the scope of the discussion to the specific experimental context would strengthen the manuscript's coherence and rigor.

Response: Our conceptual framework is summarized in Fig. 8, but we agree that the central role of microglia in this LPS model should be more explicitly stated. To address this, we have revised the Fig. 8 legend to include the following clarification and supporting references: "Microglia express higher basal levels of TLR4 than astrocytes and therefore respond earlier and more robustly to LPS. In contrast, astrocytic activation requires "priming" by cytokines released from activated microglia, making microglia the predominant initiators of the inflammatory cascade."

As addressed in an earlier response to Reviewer #2, all discussion regarding the relevance of this study to Alzheimer's disease has been removed. The aging data originally presented in Fig. 1, along with the associated discussion, have also been removed. In the revised manuscript, we now provide a more detailed discussion of the relevance of lipocalin-2 to PD (page 16). In addition, our study specifically focuses on dopaminergic neurons, microglia, and astrocytes within the substantia nigra, a brain region directly implicated in PD.

We did not assess neurodegeneration in this LPS model because we intentionally selected a paradigm that does *not* produce rapid neuronal loss, thereby avoiding potential confounding effects on the mechanisms under investigation. As discussed above, prior studies using the same LPS regimen have shown that microglial activation emerges within hours of peripheral LPS injection, but dopaminergic neurodegeneration in the substantia nigra develops much later, approximately seven months to observe modest loss (~23%) and ten months to detect more pronounced degeneration (39–47%). Thus, this model reflects an early, pre-degenerative stage of neuroinflammation. We believe this model represents an early stage of neuroinflammation preceding cell death in PD. This is critical to detect early translocation of DRP1 to the nucleus to initiate neuroinflammation. Additionally, we found that this paradigm markedly increased pro-inflammatory cytokines in the VMB, yet mitochondrial respiration in microglia remained intact. This justification and relevant references have been elaborated more in Discussion (page 16-17).

2. Please clarify the rationale for using partial instead of full knockout Dnm1I models. In this regard, a deeper characterization of the employed mutant mice with heterozygous Dnm1I knockout is needed. From the molecular perspective, it would be important to analyze Dnm1I expression levels, both at the mRNA and protein level in ventral midbrain extracts, as well as the phosphorylation status of DRP1. Additionally, the expression of key components of the NFκB pathway (such as IKK activation, and the protein levels of IκBα, p50, and p65) should be assessed in comparison to wild-type controls. Related

to mitochondrial physiology, it would be crucial to describe the impact of deleting one copy of *Dnm1* gene on mitochondrial activity in the brain.

Response: Please note that we extensively characterized the constitutive *Dnm1*^{+/-} mouse model in our previous publication (PMID: 38504290), which was referenced in this study. Briefly, we used heterozygous instead of homozygous *DRP1*-KO mice because complete deletion of *Dnm1* is embryonically lethal. Quantitative analysis using qPCR and immunoblotting in multiple brain regions confirmed an approximately 50% reduction of *DRP1* levels in the *Dnm1*^{+/-} mice. These mutant mice are indistinguishable from their WT

littermates in physical appearance, motor and cognitive function. Stereological cell counting shows comparable numbers of DA neurons in the substantia nigra pars compacta (SNpc). Assessments of mitochondrial morphology and function in adult mice show no genotype-dependent differences. Together, these data indicate that deleting one copy of *Dnm1* in the heterozygous mice does not negatively impact mitochondria, animal development and neuronal viability. However, as requested by this reviewer, we performed additional experiments to characterize this mouse model. As seen below, RNAseq data show other than *Dnm1*, all other genes are not affected in the *Dnm1*^{+/-} mice (**Supplementary Fig. 10**). Consistent with these results, no genotypic differences

between *Dnm1*^{+/-} and *Dnm1*^{+/+} littermates were detectable for the following analyses: qPCR for mitochondrial dynamics regulators (**Supplementary Fig. 11**), immunoblotting for phosphorylation of *DRP1*-616 and *DRP1*-637 - the two major post-translational modifications (**Supplementary Fig. 12**), qPCR for *Nfkbia* and *Rela* transcripts as well as immunoblotting for their respective proteins I κ B α and NF κ B-p65 (**Supplementary Fig. 13**). In combination, these results indicate that *Dnm1*^{+/-} mice were indistinguishable from their WT littermates. These results have now been included as Supplemental Information and discussed on page 19.

Supplementary Fig. 12. Protein levels of phosphorylated DRP1-S616 and DRP1-S637 in *Dnm1*^{-/-} and *Dnm1*^{+/+} mice. Microdissected ventral midbrains from *Dnm1*^{+/+} and *Dnm1*^{-/-} mice was processed for SDS-PAGE immunoblotting to assess phosphorylation of DRP1 at Ser616 (a) and Ser637 (c). Interesting, pDRP1-S637 migrates faster than pDRP1-S616 in mouse brain lysates. Total protein staining (bottom panels) was used as the loading control. **b**, Quantification of pDRP1-S616 levels from (a). **d**, Quantification of pDRP1-S637 levels from (c). *N* = 5–6 mice per group (2–4F and 2–4M); unpaired *t*-tests.

Supplementary Fig. 13. mRNA and protein levels of *NfkbialkBa* and *Relap65* in *Dnm1*^{-/-} and *Dnm1*^{+/+} mice.

a, NanoString nCounter analysis of *NfkbialkBa* and *Relap65* transcript levels.
b–c, Immunoblot analysis of *IkBa* and corresponding quantification.
d–e, Immunoblot analysis of NF- κ B p65 and corresponding quantification.
N = 6–7 mice per group (2–3 F and 3–4 M), unpaired *t*-tests.

Specific comments:

Figure 1. To better illustrate the protective role of *Dnm1*, please provide a more comprehensive description of all dysregulated transcripts comparing LPS-*Dnm1* +/- vs LPS-*Dnm1* +/+, not just the top ten or the table. Include those validated at the protein level (e.g., IL-6, IL-1 β , TNF α) in next figures.

Response: A Volcano plot of a total 757 genes showing the dysregulated transcripts induced by LPS as shown has been added to the revised Fig. 1 in the manuscript. The increases in *Lcn2*, *Cxcl10*, and *Ccl2* were validated at protein levels. Interestingly, additional proteins were detectable but not their transcripts. We believe this is due to key differences between transcript-level detection (Nanostring) and Protein-level detection (MSD multiplex ELISA immunoassay). For example, mRNA can be degraded rapidly while the protein remains stable. Nanostring is highly specific but it has a narrow dynamic range of mRNA abundance whereas MSD uses electrochemiluminescence and can detect very low femtogram-levels protein

concentrations. Last, there are temporal differences in gene vs protein expression. Changes in mRNA often occur earlier and transiently whereas proteins may accumulate or persist longer. This discussion has now been added to page 7.

Figure S3. To rule out mitochondrial misbalance is crucial to include positive control (e.g., rotenone or paraquat) to validate OCR respiration measures. Also, autophagy should be analyzed, especially considering previous work by the same group showing that partial Dnm1 knockout improves autophagic flux independently of mitochondrial function.

Response: Using paraquat as a positive control for mitochondrial respiration, we treated the neuronal N27 cells with paraquat, which reduced both basal and spare respiratory capacity. These results have now

Supplementary Fig. 4. Mitochondrial respiration in PQ-treated mouse primary microglia. Cells were treated with 100 μ M PQ for 24 h before undergoing a mito-stress assay using the XFe96 Extracellular Flux Analyzer. **a**, Representative kinetic traces from the mito-stress assay. **b**, Quantification of oxygen consumption rate (OCR) following PQ treatment. N = 3 independent experiments, each with 6–8 technical replicates; unpaired t-tests.

been added to **Supplementary Fig. 4 (panels i & j)**.

As requested, we assessed autophagy in LPS-treated primary microglia from *Dnm1*^{+/+} and *Dnm1*^{+/-} pups. Our data show that autophagy impairment induced by LPS was protected by partial loss of DRP1 function (**Supplementary Fig. 9**). This protection is consistent with our previous study. A discussion of these data has been included on page 9.

Figure 6. Regarding panel 6e, why are DRP1 nuclear levels similar between *Dnm1^{f+/+}* and saline-treated controls? There is a reduction only in the cytosolic fraction. This analysis should be repeated in whole-cell extracts to assess total protein levels (this is closely related to the second general comment).

Response: As mentioned above, we quantified the total protein levels of DRP1 in the *Dnm1^{f+/+}* mice in a previous study using whole cell extracts. We confirmed this expression level in the present study

Supplementary Fig. 6. DRP1 levels in microglial whole cell lysates.

Primary microglia from *Dnm1^{f+/+}* and *Dnm1^{f-/-}* mice were treated with 100 ng/mL LPS for 6 h, followed by whole-cell protein extraction using RIPA buffer.

a, Immunoblot analysis of DRP1 (top panel). Total protein staining (bottom panel) served as the loading control.

b, Quantification of DRP1 levels shown in **(a)**. $N = 4$ independent experiments; two-way ANOVA followed by Tukey's post hoc test.

also using whole cell extracts as suggested by the reviewer. These results have now been added to **Supplementary Fig. 6**. We believe reducing total level of DRP1 by about 50% doesn't necessarily translate into reduction of nuclear levels since the correlation might not be linear and only a fraction of DRP1 is in the nucleus.

Figure S4. An explanation addressing why Histone H3 levels are reduced in nuclear extracts from mutant mice compared to wild-type controls could be added.

Response: Thank you for pointing this out. To determine whether Histone H3 levels were truly reduced, we increased the sample size (n = 4 instead of the original n

= 2) and quantified H3 levels in the nuclear fractions. The results show no significant differences across genotypes or treatment conditions. We have now replaced the original plot with a more representative one in the revised Supplementary Fig. 5.

Supplementary Fig. 5 No differences were detected in the Histone H3 protein levels. *Dnm1*^{+/+} and *Dnm1*^{+/-} primary microglia were treated with 100 ng/ml LPS for 6h followed by nuclear and cytoplasmic fractionation. Immunoblotting of Histone H3 in the cytoplasmic and nuclear fractions, and total proteins per lane were used as loading control (Left panels). The quantification for the levels of Histone H3 (Right panel).

Figures 6 and 7. Since LPS can activate NFκB independently of DRP1, it is crucial to show that the identified DRP1 motif is functional. Therefore, the proposed DRP1 nuclear translocation and binding to Relα regulation sites could be strengthened by: Showing p65 protein levels in both cytosol and nucleus (Figure 6e–f), highlighting the sequence of the DRP1 motif identified by P7 primers (Figure 7), demonstrating that DRP1 binding leads to increased Relα transcription, to support the claim made in line 327.

Response: To address the reviewer's requests, we conducted a series of additional experiments, the results of which are now presented in Fig. 7. First, to further validate DRP1's transcriptional regulation of the *Relα* promoter, we cloned the 255 bp DRP1-binding region identified by our ChIP assay (Fig. 7a–b) into the pGL4.20[luc2/Puro] reporter vector (Fig. 7c). Second, we transfected this construct, along with a promoterless control vector, into mouse neuronal cells in the presence of either *Dnm1*-specific siRNA or a scrambled control. As shown in Fig. 7d, luciferase activity driven by the DRP1-binding region of the *Relα* promoter was markedly increased ($p < 0.0001$), whereas the promoterless vector showed no activity. Moreover, *Dnm1* siRNA-mediated knockdown significantly reduced luciferase activity compared with the scrambled control ($p = 0.0048$; Fig. 7d). The efficiency of *Dnm1* knockdown was $71.21 \pm 0.68\%$, as determined by qRT-PCR (**Supplementary Fig. 7**). These results collectively demonstrate that DRP1 regulates the *Relα* promoter through a transcriptionally active mechanism. Third, we performed a time-course analysis to determine the temporal sequence of DRP1 and NF-κB p65 nuclear accumulation following LPS stimulation. As shown in Fig. 7g–i, nuclear DRP1 levels increased as early as 0.5 h after LPS exposure, followed by an increase in nuclear NF-κB p65 beginning at 3 h, indicating that DRP1 translocates to the nucleus prior to NF-κB p65. Notably, we did not observe a significant increase in cytosolic NF-κB p65 levels up to 6 h after LPS treatment, supporting the notion that DRP1 promotes NF-κB p65 nuclear translocation rather than retention in the cytosol.

Fig. 7. DRP1 enrichment at the *Rela* promoter upon LPS exposure is transcriptionally active.

a, Schematic of the ~2000 bp *Rela* promoter region containing seven primer sets (P1–P7) used for ChIP analysis. DRP1 binding was enriched at a ~255 bp region proximal to the *Rela* transcription start site following LPS treatment.

b, Primary WT microglia were treated with 100 ng/mL LPS or vehicle for 6 h, and ChIP assays were performed to assess DRP1 enrichment at the *Rela* promoter. “% input” values represent mean ± SEM from four independent experiments, two-way ANOVA with Tukey’s multiple-comparison post hoc test.

c, Schematic of the pGL4.20[*luc2*/Puro]-*Rela* luciferase reporter containing the DRP1-binding region of the mouse *Rela* promoter, along with the promoterless pGL4.20[*luc2*/Puro] empty vector control.

d, HT22 neuronal cells were transfected with either the pGL4.20[*luc2*/Puro]-*Rela* construct or empty vector for 24 h, in the presence or absence of *Dnm1l* siRNA or scramble control. Untransfected cells were included as an additional control. Luciferase activity was measured 48 h post-transfection.

e–i, Primary WT microglia were treated with vehicle or 100 ng/mL LPS for 0.5, 1, 3, or 6 h. Immunoblot analysis of DRP1 (**e**) and NF-κB p65 (**f**) was performed in nuclear and cytoplasmic fractions (top panels). Total protein staining (bottom panels) served as loading control. **g**, quantification of nuclear and cytoplasmic DRP1 from (**e**). **h**, quantification of nuclear and cytoplasmic NF-κB p65 from (**f**). **i**, temporal profiles of DRP1 and NF-κB p65 enrichment in nuclear and cytosolic fractions demonstrate that DRP1 nuclear translocation precedes that of NF-κB p65. *N* = 4 independent experiments, one-way ANOVA with Tukey’s post hoc test.

Minor Points:

Line 38: “Low-grade chronic inflammation” better describes the inflammatory state in neurodegenerative diseases.

Response: We have changed to “chronic inflammation” as suggested but refrained from stating the grade level of inflammation.

Line 65: Please elaborate on the neurotoxic role of Lipocalin-2.

Response: Please see the response to reviewer #2 above

Line 71: Use “levels are” instead of “levels is”.

Response: Revised as suggested.

Line 119: Replace “upregulation” with “was upregulated”.

Response: Revised as suggested.

Line 117: Clarify whether the global DRP1-KO mouse refers to a previous study (add reference) or unpublished data.

Response: A reference has been added.

Response: Line 195: The sentence appears to be cut off (please revise).

Response: Thank you for pointing this out. This sentence was out of place and has been removed.

Line 432: The sentence is split across two lines (please correct).

Response: This sentence has been corrected.

LPS (cat# L4391, Sigma-Aldrich): Include the strain type to improve reproducibility.

Response: The strain type has been added.

Supplementary Table 1: The PDF formatting is unclear (please improve the table layout for readability maybe in an Excell file if possible).

Response: Excel files have been provided.

Figure S1a: Add a scale bar to indicate magnification.

Response: A scale bar has been added.

Reviewer #4

Lai reports a novel mechanism underlying DRP1-mediated inflammation induced by LPS. Reduced DRP1 expression significantly suppresses LPS-induced inflammation, including the proinflammatory molecule lipocalin 2. Upon LPS stimulation, DRP1 translocates from the cytosol to the nucleus, where it binds to the promoter region of RelA (encoding NF- κ B), activating its gene products and downstream inflammatory cytokines. These findings highlight the important role of proinflammatory lipocalin 2 in the brain.

Together, this study reveals a previously unrecognized function of DRP1 in mediating neuroinflammation via the NF- κ B–lipocalin 2 axis. The experiments are well-designed and clearly presented. Only minor revisions are suggested.

Suggestions:

- Please expand the discussion on the significance and rationale for using the LPS-induced model, particularly in relation to DRP1-mediated inflammation. How well does this model reflect the in vivo mechanisms of neuroinflammation?*

Response: Systemic injection of LPS has been used to induce acute, chronic and progressive neuroinflammation resulting in loss of the nigral dopaminergic neurons in animal models of PD. As addressed to other reviewers above, this particular LPS regimen was selected because it has been reported to induce rapid and sustained microglia activation in the substantia nigra. This is critical to detect early translocation of DRP to the nucleus to initiate neuroinflammation. We believe this model represents an early stage of neuroinflammation preceding cell death in PD. However, as acknowledged in the limitations of this study, the acute nature of this LPS model may not fully capture the chronic inflammatory milieu. This discussion has now been added to page 20.

- It would also strengthen the manuscript to discuss the outcomes of additional models (if available) that explore the relationship between DRP1-mediated mitochondrial morphology changes, neuroinflammation, and neurodegeneration. This could provide further support for the role of DRP1 in disease progression and therapeutic targeting.*

Response: The following paragraph has been added to the Discussion on page 19: “Given the central role of DRP1 in cellular stress responses, these findings suggest that DRP1-mediated

transcriptional control may contribute to a wide spectrum of inflammatory and neurodegenerative diseases. Indeed, DRP1 inhibition has shown therapeutic potential across numerous models, including PD^{10, 11}, Alzheimer's disease¹²⁻¹⁴, Huntington's disease^{12, 15} and amyotrophic lateral sclerosis¹⁶, where it protects against mitochondrial fragmentation, mitochondrial function, neuroinflammation, and neurodegeneration. The present work further expands the protective repertoire of DRP1 inhibition by establishing its transcriptional role in regulating neuroinflammatory pathways.”

Once again, we sincerely appreciate the reviewers' time, constructive comments and support of this study. We hope that our responses and revisions satisfactorily address the reviewers' comments and that the revised manuscript is now suitable for publication in *Nature Communications*.